# Neonatal-derived IL-17 producing dermal γδ T cells are required to prevent spontaneous atopic dermatitis

**Nicholas A Spidale[†], Nidhi Malhotra[‡], Michela Frascoli, Katelyn Sylvia, Bing Miu, Coral Freeman, Brian D Stadinski, Eric Huseby, Joonsoo Kang[*]**

Department of Pathology, University of Massachusetts Medical School, Worcester, United States

**Abstract** Atopic Dermatitis (AD) is a T cell-mediated chronic skin disease and is associated with altered skin barrier integrity. Infants with mutations in genes involved in tissue barrier fitness are predisposed towards inflammatory diseases, but most do not develop or sustain the diseases, suggesting that there exist regulatory immune mechanisms to prevent aberrant inflammation. The absence of one single murine dermal cell type, the innate neonatal-derived IL-17 producing γδ T (Tγδ17) cells, from birth resulted in spontaneous, highly penetrant AD with many of the major hallmarks of human AD. In Tγδ17 cell-deficient mice, basal keratinocyte transcriptome was altered months in advance of AD induction. Tγδ17 cells respond to skin commensal bacteria and the fulminant disease in their absence was driven by skin commensal bacteria dysbiosis. AD in this model was characterized by highly expanded dermal αβ T clonotypes that produce the type three cytokines, IL-17 and IL-22. These results demonstrate that neonatal Tγδ17 cells are innate skin regulatory T cells that are critical for skin homeostasis, and that IL-17 has dual homeostatic and inflammatory function in the skin.

**\*For correspondence:**
joonsoo.kang@umassmed.edu

**Present address:** [†]Celsius Therapeutics, Cambridge, United States; [‡]Elstar Therapeutics, Cambridge, United States

**Competing interests:** The authors declare that no competing interests exist.

## Introduction

The incidence of atopic dermatitis (AD, eczema) is on a steep incline in industrialized nations with estimates suggesting as high as a quarter of children affected (*Shaw et al., 2011*; *Leung and Guttman-Yassky, 2014*). Clinical and genome wide association studies (GWAS) in humans reveal that dysfunction of key structural components of epidermal barrier, such as filaggrin, and hypersensitive type 2 (IL-4, IL-5, IL-9 and IL-13) and type 3 cytokine responses (IL-17 and IL-22), are contributing factors to AD onset and progression (*Irvine et al., 2011*; *Paternoster et al., 2015*; *Malhotra et al., 2016*). The contribution of skin-targeting αβ T effector cells to AD pathogenicity is largely understood from the basic focus on damaging cytokine production and inflammatory myeloid cell recruitments. It is widely accepted that aberrant skin barrier integrity and local inflammation orchestrate the activation and recruitment of type 3 cytokine producing αβ Th17/22 cells to the skin, where they are thought to be the arbiters of the major symptoms of the disease, including visible skin damage (*Koga et al., 2008*; *Suárez-Fariñas et al., 2013*; *Mirshafiey et al., 2015*; *Kobayashi et al., 2015*; *Czarnowicki et al., 2015*).

Pivotal to the establishment of coordinated skin immunity are αβ and γδ T cells, and innate lymphoid cells (ILCs). Dermal ILC2 have been shown to be critical in mobilizing type 2 cytokine responses in AD, but very little is known about the function of innate skin T cells in autoimmunity. During the neonatal period, skin is populated by several γδTCR$^+$ and αβTCR$^+$ T cell subsets, whose effector functions are thymically programmed to produce IL-17, and to a lesser extent IL-22, upon activation in tissues. IL-17 producing γδ T cells (Tγδ17) are referred to as innate-like and the γδ T cell lineage is subject to the same effector subtype classification (Types 1, 2 and 3 cytokine producers)

as adaptive T helper cells and ILCs. Tγδ17 cells expressing Vγ2TCR (Garman TCRγ nomenclature *Garman et al., 1986*) are exported from the thymus after birth and rapidly populate the newborn dermis. These cells are part of the neonatal wave of tissue-resident lymphocytes that are not generated efficiently from adult bone marrow hematopoietic stem cells (*Spidale et al., 2019*).

Studies to date have established that Vγ2$^+$ Tγδ17 cells are the central population of the skin immunocyte subsets and are the most dominant IL-17 producing cells upon acute skin inflammatory perturbations (*Cai et al., 2011*; *Naik et al., 2012*; *Malhotra et al., 2013*; *Gray et al., 2013*; *Riol-Blanco et al., 2014*). Vγ2$^+$ Tγδ17 cells are absolutely required for acute Imiquimod (TLR7-agonist)-induced psoriasis in adult mice. Humans with the loss of function allele of the IL-17R signaling component ACT1 (*TRAF3IP2*) are more susceptible to psoriasis (*Wang et al., 2013*), but Act1-deficient mice are afflicted with spontaneous skin inflammatory diseases (*Qian et al., 2004*; *Matsushima et al., 2010*). Moreover, mice that lack IL-17R on radioresistant epithelial cells develop AD, in genetic background with a type 2 cytokine production bias (*Floudas et al., 2017*). In the former, skin pathology was attributed to hyper IL-22 production, and in the latter, diminished filaggrin expression and impaired skin barrier was implicated as the cause of AD susceptibility. In both models the apparent disease-protective function of IL-17 in skin homeostasis was not addressed and the critical source of homeostatic IL-17 is unknown.

Increases in Tγδ17 cells in patients with aberrant skin inflammation have been observed in AD (*Cai et al., 2011*; *Laggner et al., 2011*; *Nestle et al., 2009*), but accurate assessments of their contribution to human disease has lagged, in part due to challenges of isolating these cells from human tissues (*Toulon et al., 2009*). Possible dual homeostatic and inflammatory roles for IL-17 and IL-22, or cells that can produce them, have also limited the use of cytokine and T cell deficient mice to unveil their context-dependent contribution to skin disease pathogenesis. We show here that mice specifically lacking Vγ2$^+$ Tγδ17 cells succumb to a highly penetrant spontaneous AD that captures most characteristic disease features of human AD. Fulminant disease in the mice is associated with hyperactive ILC2 and requires both skin commensal bacteria (CB) and expansion of clonal αβ T cells. The initial trigger for the disease is linked to aberrant keratinocyte differentiation at young ages. Thus, Vγ2$^+$ Tγδ17 cells are essential to maintain skin homeostasis, in part by promoting normal keratinocyte barrier formation in perinatal period.

## Results

### Spontaneous AD in *Sox13*$^{-/-}$ mice specifically lacking Vγ2TCR$^+$ dermal Tγδ17 cells

To study the role of Vγ2$^+$ Tγδ17 cells in skin immunity, we generated mice deficient in *Sox13*, an HMG box transcription factor (TF) essential for their development (*Malhotra et al., 2013*; *Melichar et al., 2007*). In the immune system *Sox13* expression is restricted to early hematopoietic stem/progenitors and γδ T cells. Mice lacking *Sox13* have a highly selective defect in Vγ2$^+$ Tγδ17 cell development with all other hematopoietic cell types normally preserved (*Malhotra et al., 2013*; *Gray et al., 2013*). One exception is innate iNKT17 cells that are partially affected in the lymph nodes (LNs) (*Malhotra et al., 2018*), but these cells are rare in the skin. Loss of Vγ2$^+$ T cells was also observed in the skin of *Sox13*$^{-/-}$ mice, while Vγ4$^+$ (Vγ2$^-$ TCRδ$^{int}$) T cells were present at a normal frequency and were capable of producing high levels of IL-17A (*Figure 1—figure supplement 1A*). Incompatible with the pro-inflammatory nature of Vγ2$^+$ Tγδ17 cells,>90% of *Sox13*$^{-/-}$ mice maintained on a 129/Sv genetic background (>250 mice cumulatively tracked over several years) of both sex develop visible dermatitis in the muzzle, ears, eyes and elsewhere around three to four months of age (*Figure 1—figure supplement 1B*), displaying many of the hallmarks of human AD (*Leung and Guttman-Yassky, 2014*; *Zheng et al., 2007*; *Fujita, 2013*; *Kim, 2015*). Notably, while we have previously reported perinatal lethality in *Sox13*-deficient C57BL/6 mice, no gross developmental abnormalities were observed in 129.*Sox13*$^{-/-}$ mice for >1 year despite the development of AD-like disease. Pathophysiology included epidermal thickening (acanthosis, *Figure 1A*, left), marked accumulation of immunocytes in skin epithelial lesions leading to eosinophilia, neutrophilia, and increases monocytes (Mo) and Mo-derived dendritic cells (DCs) in the skin (*Figure 1A–F*). Further, mast cells were expanded, but this trend did not reach statistical significance (*Figure 1—figure supplement 1C*). *Sox13*$^{-/-}$ mice exhibited aberrant, high frequency scratching behavior coincident

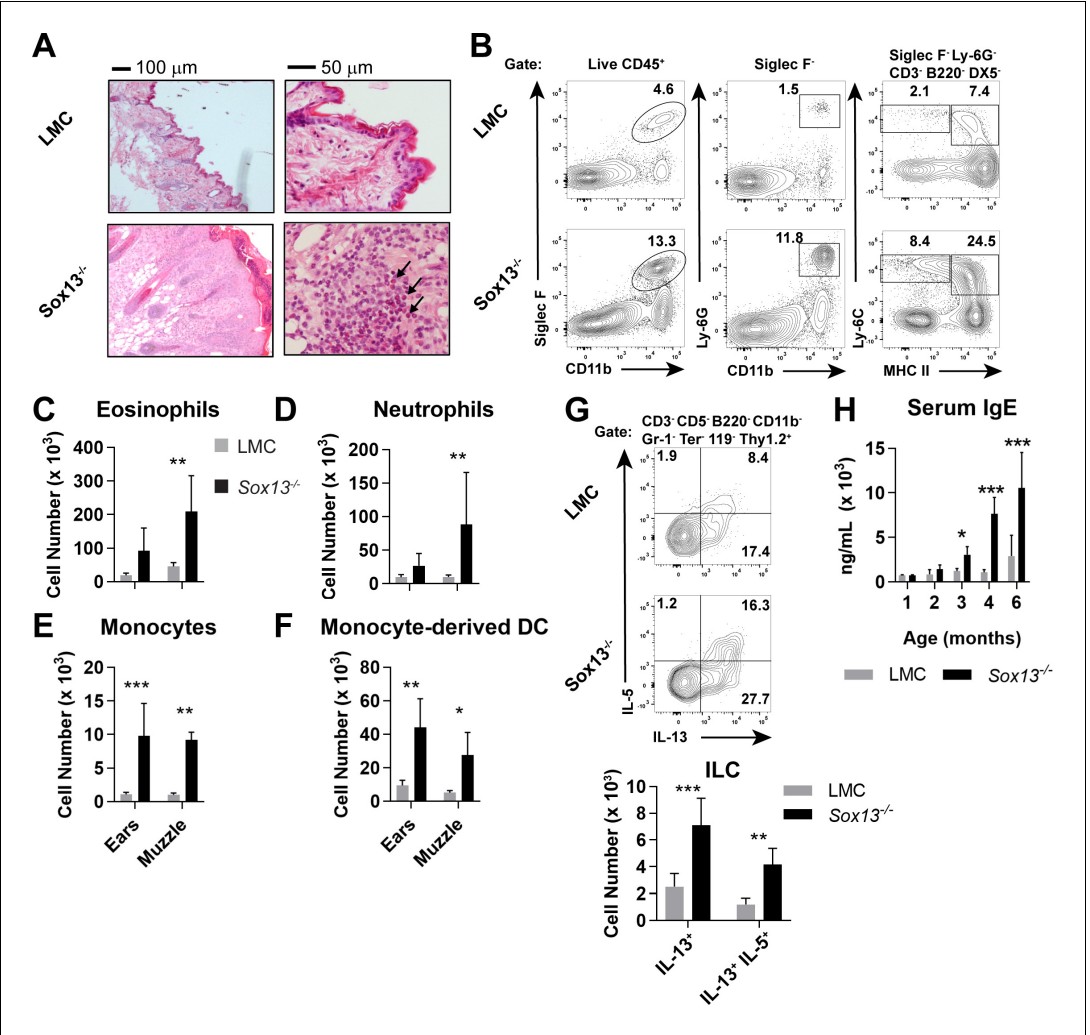

**Figure 1.** Development of AD in the absence of dermal Vγ2⁺ Tγδ17 cells. (**A**) Biopsies of muzzle skin from 6 mo *Sox13⁻/⁻* and *Sox13⁺/⁻* littermate control (LMC) was analyzed by H and E staining. Black arrows identify numerous eosinophilic infiltrates in the epidermis. Representative of four experiments, each with minimum n = 2/group. (**B**) Muzzle skin was digested and analyzed via FACS for Siglec F⁺ eosinophils (left panels), Ly- 6G⁺ neutrophils (middle panels), Ly-6C⁺ MHC-II^lo monocytes and Ly-6C⁺ MHC-II^hi monocyte-derived dendritic cells (right panels). Data are representative of >6 similar experiments analyzing 2–3 mice per/group. (**C–F**) Enumeration of cell types examined in Panel B. n = 6/group. *, p<0.05; **p<0.01; ***, p<0.001 by ANOVA. (**G**) Muzzle-infiltrating cells were isolated from LMC and *Sox13⁻/⁻* mice and re-stimulated in vitro with PdBu/ionomycin to assess production of IL-5 and IL-13 by ILCs. ILC identified as CD45⁺ Thy1.2⁺Lineage markers^neg (CD3/CD4/CD5/CD8/CD11b/DX5/Gr-1/ TCRδ/TCRβ/Ter-119^neg). Bottom summary graph enumerates IL-13⁺ and IL-5⁺IL-13⁺ ILC. N = 6/group. **p<0.01; ***p<0.001 by ANOVA. (**H**) Serum IgE concentration in mice of indicated genotype, aged 1–6 mo, was determined by ELISA. n = 3–6/group. *, p<0.05; ***, p<0.001 by ANOVA.

The online version of this article includes the following figure supplement(s) for figure 1:

**Figure supplement 1.** Specific loss of Vγ2⁺ Tγδ17 cells, scratching behaviors and reciprocally enhanced effector function of ILCs in *Sox13⁻/⁻* mice with dermatitis.

with visible skin lesions (*Figure 1—figure supplement 1D* and *Videos 1* and *2*), suggesting an enhanced itch response. In addition, expanded ILC2 (GATA3^hi) associated with human AD (*Kim, 2015*; *Salimi et al., 2013*; *Roediger et al., 2014*), and their capacity to produce the type 2 cytokines IL-5 and/or IL-13, was recapitulated in *Sox13⁻/⁻* mice (*Figure 1G*, *Figure 1—figure supplement 1E–G*). Conversely, in young *Rora⁻/⁻* mice lacking in ILC2 (*Wong et al., 2012*) there is an increase in Vγ2⁺ Tγδ17 cells with enhanced capacity to produce type 3 cytokines (*Figure 1—figure*

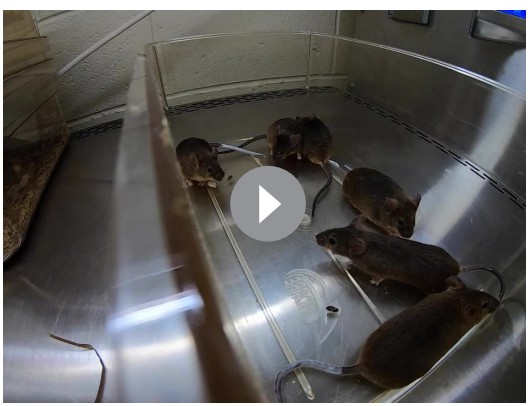

**Video 1.** Comparative Scratching behavior of *Sox13*[-/-] and WT control (tails painted solid white) mice.
https://elifesciences.org/articles/51188#video1

*supplement 1H*), suggesting a possible counter-regulation between Tγδ17 cells and ILC2. Crucially, age-dependent increases in IgE titer, evident by 3 months of age of the mice (*Figure 1H*), before visible signs of disease, captured one of the major symptoms of human AD.

## Expanded αβ T cells are required for AD

Significant expansion of αβ T cells in the skin of *Sox13*[-/-] mice was evident starting ~3 months of age, prior to any visible skin inflammation. Both CD4[+] and CD8[+] T cells increased in numbers up to 10-fold by 6 months of age, depending on skin sites (*Figure 2A*). Notably, CD4[neg]CD8[neg] (double negative, DN) T cells accounted for 10–20% of TCRβ[+] cells in the skin of both LMC and *Sox13*[-/-] mice, with a significant expansion observed in *Sox13*[-/-] skin (*Figure 2A*). Utilizing the MR1/5-OP-RU tetramer, we identified that the DN subset in both healthy and AD skin consisted primarily of MAITs (*Figure 2B*). CD4[+] or CD8[+] MAITs were rare in the skin of WT mice, with only marginal increase in CD8[+] MAITs in *Sox13*[-/-] skin (*Figure 2—figure supplement 1A*). In the skin draining LNs (dLNs), only subtle increase in the frequency of MAITs was observed in *Sox13*[-/-] mice, with the majority being the CCR6[+]CD4[-]CD8[-] subset in all mice (*Figure 2—figure supplement 1B–C*). iNKT cells were rare in the skin and no significant alterations were observed in *Sox13*[-/-] mice (*Figure 2—figure supplement 1D*).

The majority of αβ T cell subsets in AD were associated with enhanced capacity to produce both IL-17 and IL-22, whereas in control mice very few CD4[+] or CD8[+] αβ T cells were capable of IL-17 production, and even more constrained IL-22 secretion was evident (*Figure 2D*). DN MAIT cells were primed for IL-17 in both LMC and *Sox13*[-/-] mice. In contrast to the enhanced type 3 cytokine production, the frequency of Th2 cells was not altered significantly in *Sox13*[-/-] skin, although numerically they were also increased. Similarly, although the frequency of skin FOXP3[+] regulatory T cells (Tregs) was decreased in the ear (*Figure 2—figure supplement 1E*), but not muzzle, of *Sox13*[-/-] mice, their numbers were comparable to controls, indicating preferential expansion of effector populations. Matching the T cell expansion in skin there was an ~8 fold expansion in cellularity in dLNs (*Figure 2E*). The trend to this increase was evident before visible skin lesions, at ~3 months of age, and was associated with greatly increased numbers of spontaneous germinal centers (GCs), typical of autoimmune disorders (*Domeier et al., 2017*), with aberrant GC formation (green, *Figure 2F*) in the T cell zone (blue, *Figure 2F*) and increased number of follicular T help (Tfh) cells, GC B cells and plasma cells (*Figure 2G* and *Figure 2—figure supplement 1F,G*). To ascertain changes in the expression of secreted inflammatory mediators, RNA was isolated from the muzzle skin at 6 months of age and select cytokine and chemokine gene expression was assessed by quantitative RT-PCR (*Figure 2—figure supplement 1H*). A coordinate induction of the cytokines IL-1β, IL-6 and IL-23, which promote type 3 cytokine producing lymphocytes, was prominent. A simultaneous increase in the danger associated molecular pattern molecule IL-33 was observed, which has been associated with skin inflammation and itch response (*Salimi et al., 2013*; *Meephansan et al., 2013*).

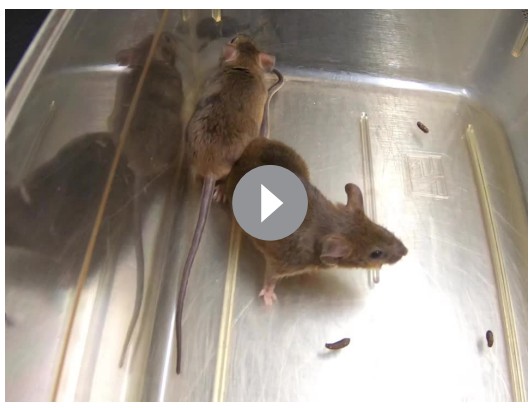

**Video 2.** Isolated scratching episode typical of *Sox13*[-/-] mice.
https://elifesciences.org/articles/51188#video2

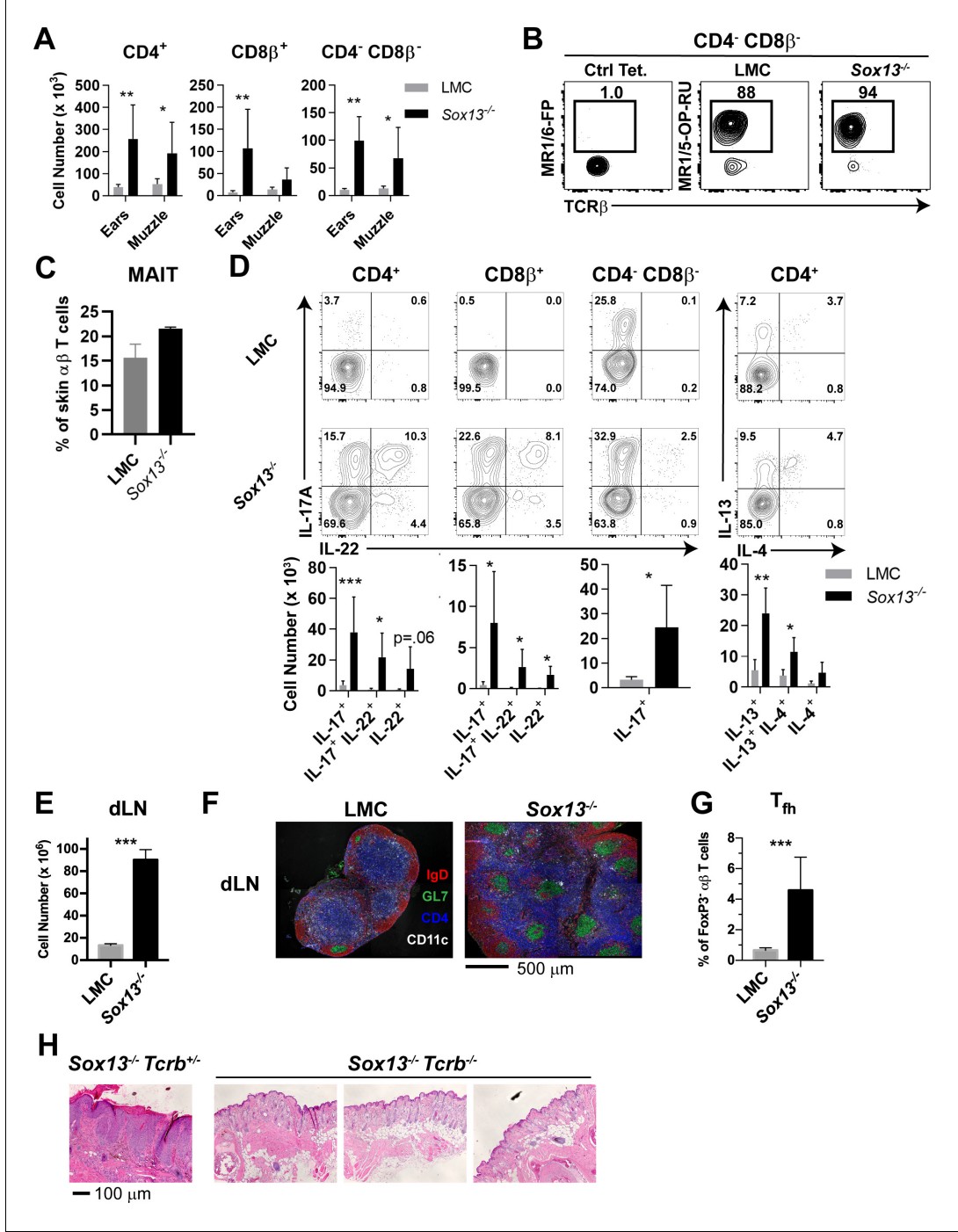

**Figure 2.** Aberrant αβ T cell activation in AD of *Sox13*[-/-] mice. (A) Total number of the indicated T cell types recovered from skin of from 5-6mo mice were calculated using AccuCheck counting beads. n = 6/group. *, p<0.05; **, p<0.01 by ANOVA. (B) FACS analysis of CD4[neg]CD8β[neg] skin T cells (gated on B220[-]F4/80[-] TCRβ[+]) with control MR1/6-FP or MR1/5-OP-RU tetramer to identify MAIT cells in 5 mo mice. (C) Summary data of frequency of MAIT tetramer-reactive cells among total TCRβ[+] cells pooled from two independent experiments, performed as in Panel B analyzing a total of 5–6 mice/group. (D) Muzzle-infiltrating cells were isolated from indicated mice, stimulated in vitro with PdBu/ionomycin, and analyzed for αβ T cell subset-specific production of IL-17A and IL-22 and for CD4[+] T cell production of IL-4, and IL-13. FACS data are representative of >5 experiments. For summary data below, n = 6/group. *, p<0.05; **p<0.01, ***, p<0.001 by ANOVA or *t*-test (CD4[-]CD8β[-] cells). (E) Total cell number enumeration in skin draining LNs (dLNs) of 6 mo mice of indicated genotype, n = 6/group. ***, p<0.001 by Student's *t*-test. (F) Muzzle draining mandibular LN (dLN) from 5 to 6 mo mice were fixed in paraformaldehyde,

*Figure 2 continued on next page*

*Figure 2 continued*

frozen in OCT compound, cryosectioned, and then labeled with the indicated antibodies to visualize B cell follicles (IgD[+]), T cell zones (CD4[+]), dendritic cells (CD11c[+]), and germinal centers (GL7[+] IgD[-]). Images are representative of two experiments analyzing sections from at least 3 mice per experiment. (G) Summary data of T follicular helper (Tfh) cells in dLN of 6 mo LMC and *Sox13[-/-]* mice. Tfh cells were identified as CD4[+]FoxP3[neg] PD-1[hi] CXCR5[+]Bcl6[+]. n = 7–8/group. \*\*\*, p<0.001 by Student's *t*- test. (H) *Sox13[-/-]* and 129.*Tcrb[-/-]* mice were crossed to generate double-deficient mice, and then disease progression tracked by phenotyping and muzzle inflammation assessed by H and E staining. *Sox13[-/-]Tcrb[-/-]* mice do not develop overt or histological signs of AD at 6 mo. Data are representative of 10–15 mice of each genotype analyzed.

The online version of this article includes the following figure supplement(s) for figure 2:

**Figure supplement 1.** Characterization of skin and dLN MAITs, iNKT cells, regulatory T cells (Tregs), B cell subsets and melanocyte antigen-specific T cells in *Sox13[-/-]* mice.

---

To determine whether the expansion of skin T cells was correlated to more efficient display of skin antigens in dLN, melanocyte-specific antigen presentation in *Sox13[-/-]* mice was assessed. Naïve PMEL17 CD8[+] TCR transgenic T cells specific for a melanocyte antigen (*Harris et al., 2012*) were labeled with CellTrace Violet and transferred into *Sox13[-/-]* and WT hosts, and their proliferation was analyzed by dye dilution (*Figure 2—figure supplement 1I,J*). We observed a 3-fold increase in PMEL17 T cells proliferation in skin dLNs of *Sox13[-/-]* mice compared to controls. In contrast, no differences in proliferation were observed at distal sites, including the spleen. Finally, to demonstrate that αβ T cells are required for AD in *Sox13[-/-]* mice, skin pathology in *Sox13[-/-]Tcrb[-/-]* was monitored. The absence of αβ T cells prevented AD development with no visible evidence of skin inflammation and skin histology was grossly normal, including lack of epidermal hyperplasia (*Figure 2H*).

Collectively, these results indicated that prior to the onset of visible diseases, B and T cells expand, with evidence for IgE hyperproduction. With the progression of disease, the skin displays a prominent type 3 effector inducing cytokine milieu with attendant expansion of Th17 cells and IL-17[+] MAITs. Thus, fulminant AD in *Sox13[-/-]* mice is characterized by strong polarization and/or expansion of Th17 and Th17-like cells of αβ T cell lineage.

## Altered basal keratinocyte differentiation program in *Sox13[-/-]* mice

To map the sequence of early cellular and molecular alterations in *Sox13[-/-]* mice that can account for the eventual inflammatory immune landscape, we first assessed the impact of the loss of Vγ2[+] Tγδ17 cells on differentiating keratinocytes. For this we undertook a whole transcriptome analysis of basal CD49f[+] (*Itga6*) keratinocytes of *Sox13[-/-]* mice at 3 and 7 weeks (wks), well before the onset of aberrant skin inflammation starting in ~3 months old (mo) mice. This population was chosen because they contain keratinocyte stem cells and progenitors (*Terunuma et al., 2007*; *Sada et al., 2016*) and the two timepoints coincide with the hair follicle catagen cycle, characterized by active keratinocyte differentiation followed by the relatively quiescent telogen cycle, respectively (*Fuchs, 2007*). Notably, *Sox13* transcripts were virtually undetectable in both *Sox13[-/-]* and LMC keratinocytes, indicating that *Sox13* deficiency is unlikely to cell-autonomously impact keratinocytes. In all, 261 genes were differentially expressed (≥2 fold changes, p<0.05) between 3 wk WT vs *Sox13[-/-]* basal keratinocytes (*Figure 3A*). Gene Ontology (GO) enrichment analysis revealed pronounced cell apoptosis signatures and stress responses in *Sox13[-/-]* basal keratinocytes (*Figure 3B*). At 7wk the difference was muted with 50 genes differentially expressed (*Figure 3A*) with no significant clustering of these genes into specific biological processes, likely reflecting the resting state of basal keratinocytes in the telogen phase. Expression of only 3 genes, *Igfbp3, Mir-17hg* (Mir-17–92) and *4930480K23Rik* (non-coding RNA), was altered at both ages. *Igfbp3 and Mir-17hg* (Mir-17–92) have been shown to be associated with skin inflammations (*Edmondson et al., 2005*; *Zhang et al., 2018*) and their expression was initially decreased in *Sox13[-/-]* basal keratinocytes, but this pattern was flipped at 7wk. *Sox13[-/-]* mice prior to 2 months do not show any significant alterations in skin immune subsets or visible damage, and consistent with this *Sox13[-/-]* basal keratinocytes showed no significant alterations in the expression of inflammatory mediators of immunocytes at 3 and 7 wks. Genes encoding for the structural components of the skin barrier including gap junction proteins, extracellular matrix (except collagens at 3wk) and keratins, were also not altered in expression. However, expression of several genes critical for normal differentiation of basal keratinocytes was altered at 3wk, including

diminished expression of the IL-17 target Blimp1 (*Prdm1*) (*Magnúsdóttir et al., 2007*; *Wang et al., 2016*), *Sox9* (*Menzel-Severing et al., 2018*), *Runx1, Irf3/6, S100a11,* and increased expression of *Myc* (*Wu et al., 2015*), *Dlx3,Trp73* and *Maf*. In addition, genes in the TGFβ, Lymphotoxin and the

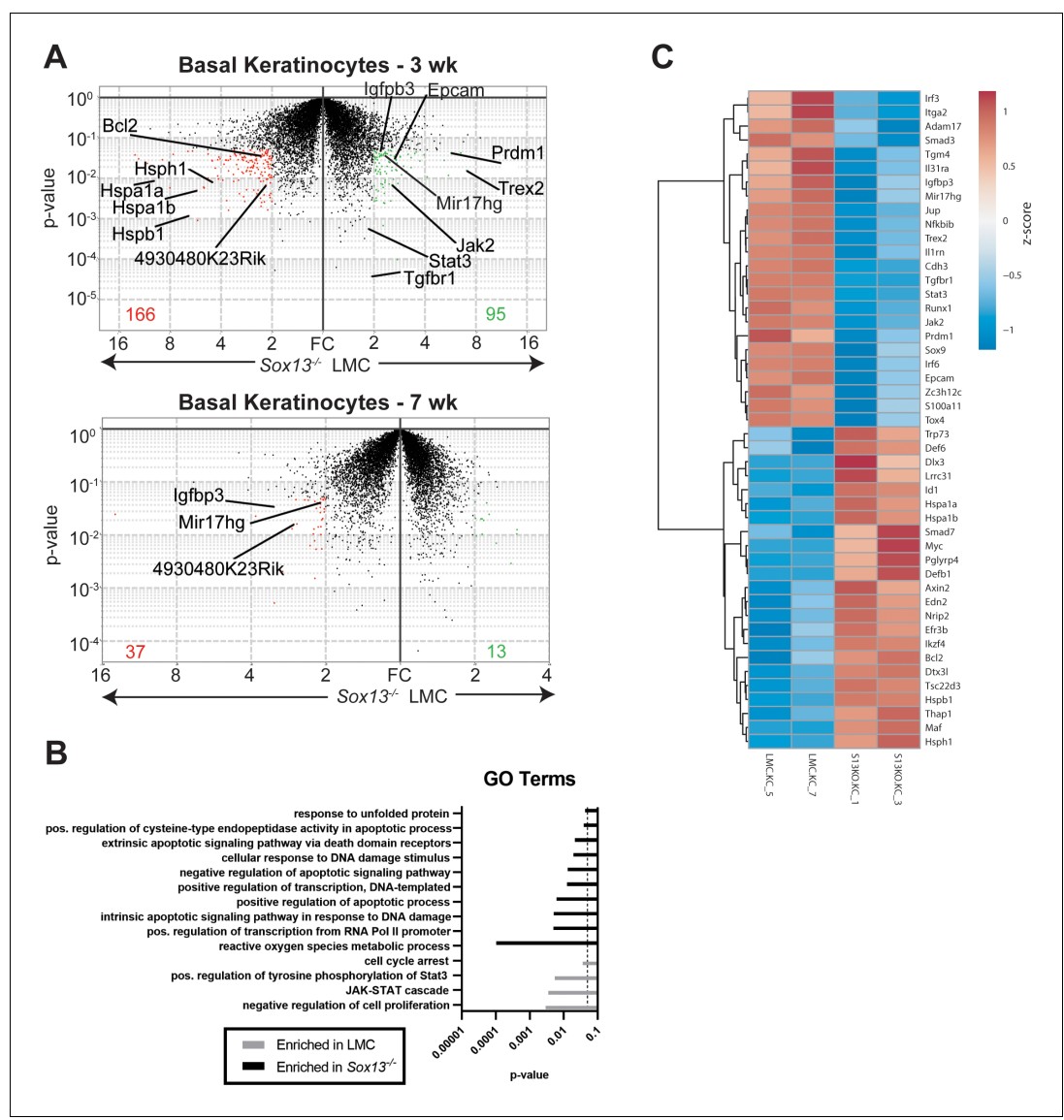

**Figure 3.** Perturbations in early basal keratinocyte transcriptome in the absence of Vγ2⁺ Tγδ17 cells. (**A**) Epidermal basal keratinocytes were sorted from 3 and 7 wk old male LMC and *Sox13⁻ᐟ⁻* mice and subjected to gene expression analysis by RNA sequencing (in biological triplicates). Red and green dots represent genes with fold change (FC) ≥2 and p-value ≤. 05 and the numbers at the bottom denote number of genes whose expression was significantly altered. Select genes are annotated. (**B**) Differentially expressed genes from Panel A were analyzed for Gene Ontology (GO) term enrichment using DAVID. Displayed are a selection of significantly enriched (p ≤. 05, dashed line) GO terms. (**C**) Heatmap of differentially expressed genes (FC ≥1.5 and p-value ≤. 05) among male 3 wk old basal keratinocytes with genes involved in cell differentiation, barrier function, skin inflammation and stress response pathways annotated.

The online version of this article includes the following source data for figure 3:

**Source data 1.** RNA sequencing read count tables used to generate volcano plot in *Figure 3A* (upper panel) for basal keratinocytes analysis of 3 wk old mice.

**Source data 2.** RNA sequencing read count tables used to generate volcano plot in *Figure 3A* (lower panel) for basal keratinocytes analysis of 7 wk old mice.

JAK-STAT signaling pathways had lower levels of expression in $Sox13^{-/-}$ basal keratinocytes. Genes controlling barrier fitness, such as *Trex2, Epcam, Adam17, Itga2, Cdh3, Tgm4, Il31ra, Il1rn* and *Jup*, were decreased in expression, whereas *Def, Lrrc31* and *Tsc22d3* (GILZ) were increased in $Sox13^{-/-}$ keratinocytes (*Figure 3C*). Together, these results indicate that Tγδ17 cells are critical for establishing normal developmental program of basal keratinocytes during the catagen cycle, and in their absence the data suggests altered keratinocyte differentiation and increased propensity to apoptosis.

## Skin commensal bacteria dysbiosis in $Sox13^{-/-}$ mice is responsible for AD

Analysis of differentiated keratinocytes at 2 months or later does not allow for clear distinction between impaired barrier function arising from keratinocyte-intrinsic defects or from inflammatory immunocyte-mediated degradation. In patients with AD, expansions of *Staphylococcus* and *Corynebacteria* species are often observed in skin lesions (*Malhotra et al., 2016*; *Kobayashi et al., 2015*; *Grice and Segre, 2011*; *Cho et al., 2010*) and mouse models of AD with barrier defects replicate the AD-associated microbiome dysbiosis. Thus, one prediction of the altered keratinocyte differentiation and barrier function well before the onset of chronic inflammation in young $Sox13^{-/-}$ mice is that the homeostasis of skin commensal bacteria (CB) with the barrier will be disrupted, with the resultant dysbiosis driving the immune responses. We tested this possibility by first establishing skin microbiota of $Sox13^{-/-}$ mice at 3 and 6 mo by 16S rRNA sequencing, followed by assessment of antibiotic treatment (Abx) on AD onset and progression. As in human AD patients, AD in $Sox13^{-/-}$ mice was associated with dysbiosis of *Staphylococcus* and *Corynebacteria,* but with distinct kinetics (*Figure 4A*). Most $Sox13^{-/-}$ mice showed an early bloom of *Corynebacteria* (*C. mastitis, Figure 4— figure supplement 1A*), with the expansion maintained in some mice, but for the majority returning to the LMC frequencies at 6 mo. Expansion of *Staphylococcus* was pronounced at the frank phase of disease but was not obvious at 3 mo. These results largely recapitulate skin CB dysbiosis in two mouse models of AD (*Kobayashi et al., 2015*; *Floudas et al., 2017*).

To determine whether skin CB is necessary for AD initiation and/or progression in $Sox13^{-/-}$ mice we treated the mice from birth or starting at 3 mo with a combination of antibiotics (cefazolin and enrofloxacin in drinking water) previously used for a similar purpose (*Kobayashi et al., 2015*). Skin commensal sequencing of Abx mice confirmed that *Staphylococcus* and *Corynebacterium* species were significantly reduced (*Figure 4—figure supplement 1B*). Regardless of regiments, the Abx $Sox13^{-/-}$ mice were protected from AD. All pathophysiological features of AD were absent, with resolution of acanthosis (*Figure 4B*), decreased serum IgE concentrations (*Figure 4C*), and suppression of myeloid expansion (*Figure 4D,E*). While CD4$^+$ cells remained at an elevated frequency, IL-17 and IL-22 production was significantly reduced (*Figure 4F*). Further, all disease-associated phenotypes of the dLN were corrected by Abx treatment, leading to reduction of total cell number, and the normalization of Tfh, GC B cell, and plasma cell frequencies (*Figure 4G–J*, *Figure 4—figure supplement 1C*). We also tested whether the disease initiation is restricted to a narrow developmental window spanning neonatal-juvenile stages. For this, $Sox13^{-/-}$ mice were treated with the antibiotic cocktail from birth and then the treatment was terminated at 3 wks of age. AD development was not prevented in mice treated only acutely at birth (data not shown), suggesting that continuous skin commensal-immunocyte crosstalk contributes to the disease postnatally and delayed/altered commensal interactions during neonatal stage do not permanently remodel skin pathophysiology.

## Tγδ17 cells respond to skin CB by IL-1 and IL-23 secreted by APCs

Commensal dysbiosis is known to result from impaired barrier functions. That Tγδ17 cells themselves normally respond to *Corynebacteria/Staphylococcus* and the absence of Vγ2$^+$ Tγδ17 cells also directly contributes to the aberrant microbiome expansion was assessed next. A recent report of Tγδ17 cell activation in SPF mice topically colonized with *C. accolens* (*Ridaura et al., 2018*) strongly supported this possibility. There are two Tγδ17 subsets in mice. Along with Vγ2$^+$ Tγδ17 cells, the dermis contains the canonical Vγ4TCR$^+$ fetal derived Tγδ17 cells, which are not dependent on *Sox13* for populating the skin (*Malhotra et al., 2013*). Thus, an obvious question is why dermal Vγ2$^+$ Tγδ17 cells are functionally non-redundant in suppressing AD initiation. Whereas Vγ4$^+$ Tγδ17 cell persistence is dependent on CB (*Duan et al., 2010*) and parallels dermal Th17 and Tc17 cells (*Naik et al.,*

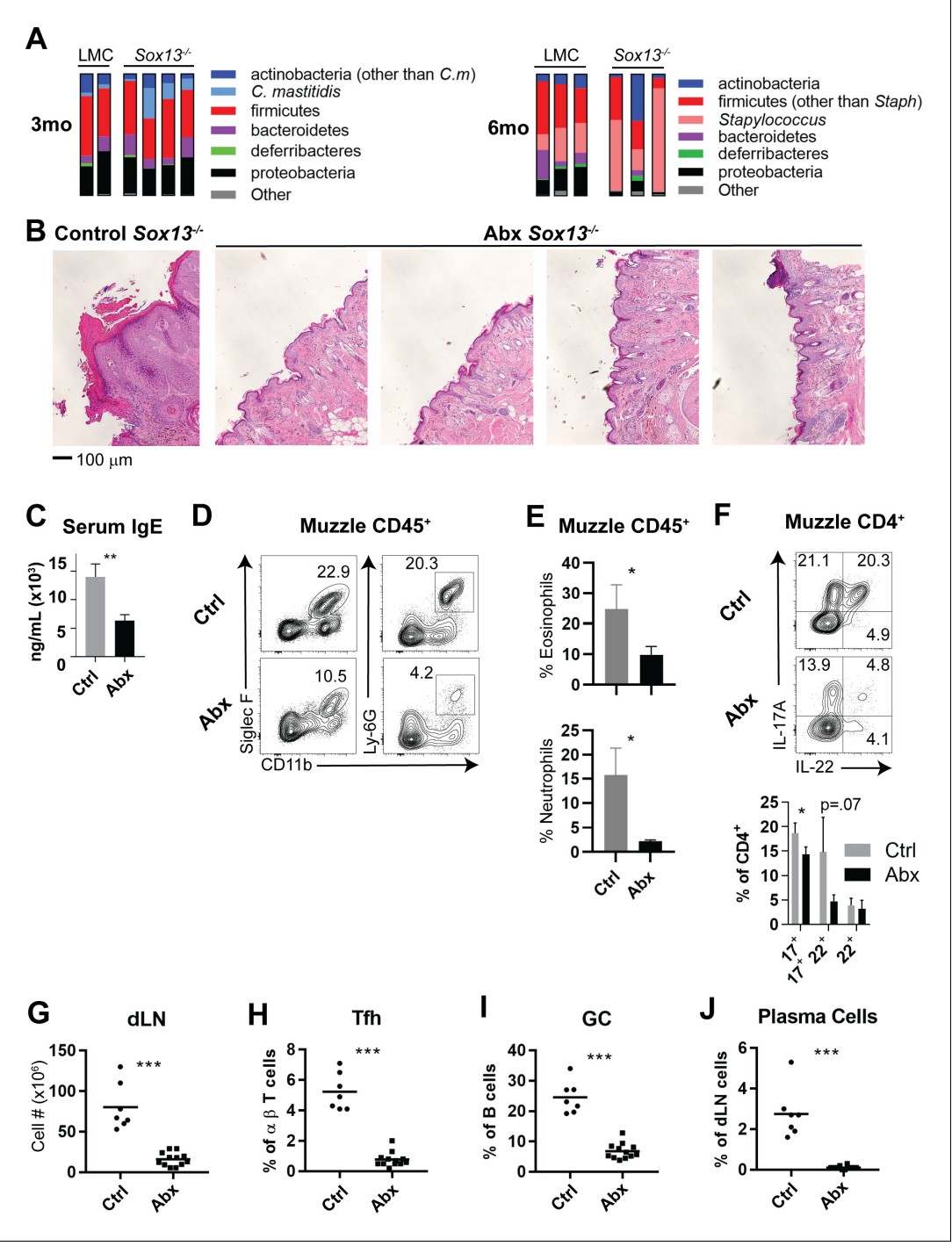

**Figure 4.** Skin commensal alterations in the absence of Vγ2+ Tγδ17 cells drive AD. (**A**) Summary stacked bar charts of muzzle skin microbiome analysis of *Sox13-/-* and LMC mice at 3 mo and 6 mo. Species depicted are annotated on the right and their corresponding frequencies among total 16S rRNA sequences are shown. One experiment of three shown. (**B**) *Sox13-/-* mice were antibiotic (enrofloxacin and cefazolin) treated (Abx) by drinking water from 2 mo and then muzzle histology analyzed at 6 mo. Images are representative of 4 analyzed Abx-treated mice, with at least 2 sections separated by >100 microns analyzed for each mouse. (**C**) Serum IgE levels of Ctrl and Abx-treated *Sox13-/-* mice at 6 mo were assessed by ELISA. n = 6 (Ctrl) or 10 (Abx). **, p<0.01 by Student's *t*-test. (**D**) Muzzle skin of 6 mo Ctrl and Abx *Sox13-/-* mice was analyzed for eosinophil and neutrophil infiltration via FACS. Data are representative of 9 analyzed Abx-treated mice from 3 independent cohorts. (**E**) Summary data of the frequency of Eosinophils (top) and Neutrophils (bottom) among all CD45+ muzzle skin cells. n = 3/group from 1 of 3 similar experiments. *, p<0.05 by t-test. (**F**) Muzzle skin of 6 mo Ctrl and Abx *Sox13-/-* mice was analyzed for
*Figure 4 continued on next page*

*Figure 4 continued*
Th17 cytokine production post PMA/ionophore reactivation. Summary data of n = 5/group, pooled from 2 independent experiments. *, p<0.05 by ANOVA. (G–J) Mandibular and parotid dLN cells from Ctrl and Abx *Sox13*[-/-] mice were analyzed for total cell number (G), and the frequency of Tfh cells (H), GC B cells (I), and CD138+ plasma cells (J). n = 7–12/group pooled from 4 independent cohorts. ***, p<0.001 by Student's *t*-test. The online version of this article includes the following source data and figure supplement(s) for figure 4:

**Source data 1.** Bacterial species abundance from the muzzle skin of 3 mo and 6 mo LMC and *Sox13*[-/-] mice as summarized in *Figure 4A*.
**Figure supplement 1.** Skin microbiome of antibiotic-treated (Abx) *Sox13*[-/-] mice and independence of dermal Vγ2+ Tγδ17 from CB for their development and maintenance.
**Figure supplement 1—source data 1.** Bacterial species abundance from the muzzle skin of 3 mo and 6 mo LMC and *Sox13*[-/-] mice treated with Abx as summarized in *Figure 4—figure supplement 1B*.

*2012*), Vγ2[+] Tγδ17 cells were not, as assessed in germ free (GF) mice (*Figure 4—figure supplement 1D*). Abx WT mice also showed the loss of skin Vγ4[+] Tγδ17 cells (Vγ2[neg]Vγ3[neg] quadrant, Supp *Figure 3E*) and the loss of tonic *Il17a* transcription by residual Vγ4[+] Tγδ17 cells (Vγ2[neg]) in Abx WT mice. In contrast, constitutive *Il17a* transcription in Vγ2[+] Tγδ17 cells was not suppressed by Abx (*Figure 4—figure supplement 1D–F*). These results indicate unique homeostatic activation requirements for dermal Vγ2[+] Tγδ17 cells.

To determine how Tγδ17 cells normally react to skin CB, γδ T cells were isolated from dLNs and stimulated with a diverse set of *Staphyloccus* and *Corynebacteria* species in transwell cultures with antigen presenting cells (APCs). While *Corynebacteria* consistently stimulated copious IL-17 but not IFNγ, production from Vγ2[+] Tγδ17 cells, so did *Staphyloccus* species, albeit with a consistent diminution of IL-17 amounts per cell (*Figure 5A*). For comparison, Vγ4[+] Tγδ17 cells showed indistinguishable pattern of CB reactivity. This Tγδ17 activation was not T-APC contact dependent, as similar levels of IL-17 production was elicited when CB-activated APCs were separated from Tγδ17 cells in transwells, indicating sufficiency of trans-acting factor(s) (*Figure 5B*). Given that IL-1 and IL-23 from activated APCs was linked to Tγδ17 effector cytokine production in peripheral tissues (*Sutton et al., 2009*), both cytokines were quenched by Ab in the same culture to test whether they are the trans activating factors in the skin. Transwell cultures in which Tγδ17 cells were cultured in a separate compartment from CB-APC and then blocked with Abs against the cytokines showed significantly reduced IL-17 production (*Figure 5C* and data not shown). Collectively, these results indicate that Vγ2[+] Tγδ17 and Vγ4[+] Tγδ17 cells respond comparably to skin CB that are altered in AD, and that this reactivity can occur independently of direct contact with CB-APCs. Thus, biases in CB recognition by Tγδ17 subsets per se are unlikely to explain the necessity of Vγ2[+] Tγδ17 cells for skin homeostasis. To date, type three cytokine producing T cells with established functions in the skin have been shown to require CB for persistence. However, Vγ2[+] Tγδ17 cells can be maintained and function in the skin independent of CB, a distinguishing characteristic that likely underpins the non-redundancy of Vγ2[+] Tγδ17 cells in controlling aberrant skin inflammation.

## Vβ4[+]vα4[+] αβ T clonotypes dominate the diseased skin of *Sox13*[-/-]mice

The expanded αβ T cells in *Sox13*[-/-] mice are required for AD progression. If the expansion is antigen driven a prediction would be that there would be restricted TCR repertoire in skin infiltrating αβ T cells of *Sox13*[-/-] mice. To test this, we first assessed TCRVβ chain repertoire of CD4[+] T cells by flow cytometry. While the TCRVβ usage of dLN T cells of WT and *Sox13*[-/-] mice was indistinguishable, skin CD4[+] T cells in *Sox13*[-/-] mice were dominated by the usage of Vβ4 TCR, starting at 3 months of age and reaching a plateau at ~5–6 months (*Figure 6—figure supplement 1A,B*). As skin inflammation progressed to overt disease (~5 mo), the frequency of Vβ4[+] CD4[+] T cells increased ~3 fold and in 5–6 mo *Sox13*[-/-] mice the total number of skin CD4[+] T cells was more than 10-fold greater in *Sox13*[-/-] mice than WT mice, depending on the skin site, with up to 50% of these cells expressing Vβ4 TCR (*Figure 6A–C*). In comparison, TCR Vβ skewing was not consistently observed for any other Vβs or for any TCRs associated with FOXP3[+] Tregs or CD8[+] T cells (*Figure 6—figure supplement 1B,C*). The increased cellularity in diseased *Sox13*[-/-] skin, combined with the strong Vβ4-bias and increased proliferation of skin Vβ4[+]CD4[+] T cells in *Sox13*[-/-] mice (*Figure 6—figure supplement 1D*), suggested that these CD4[+] T cells were undergoing expansion in the skin.

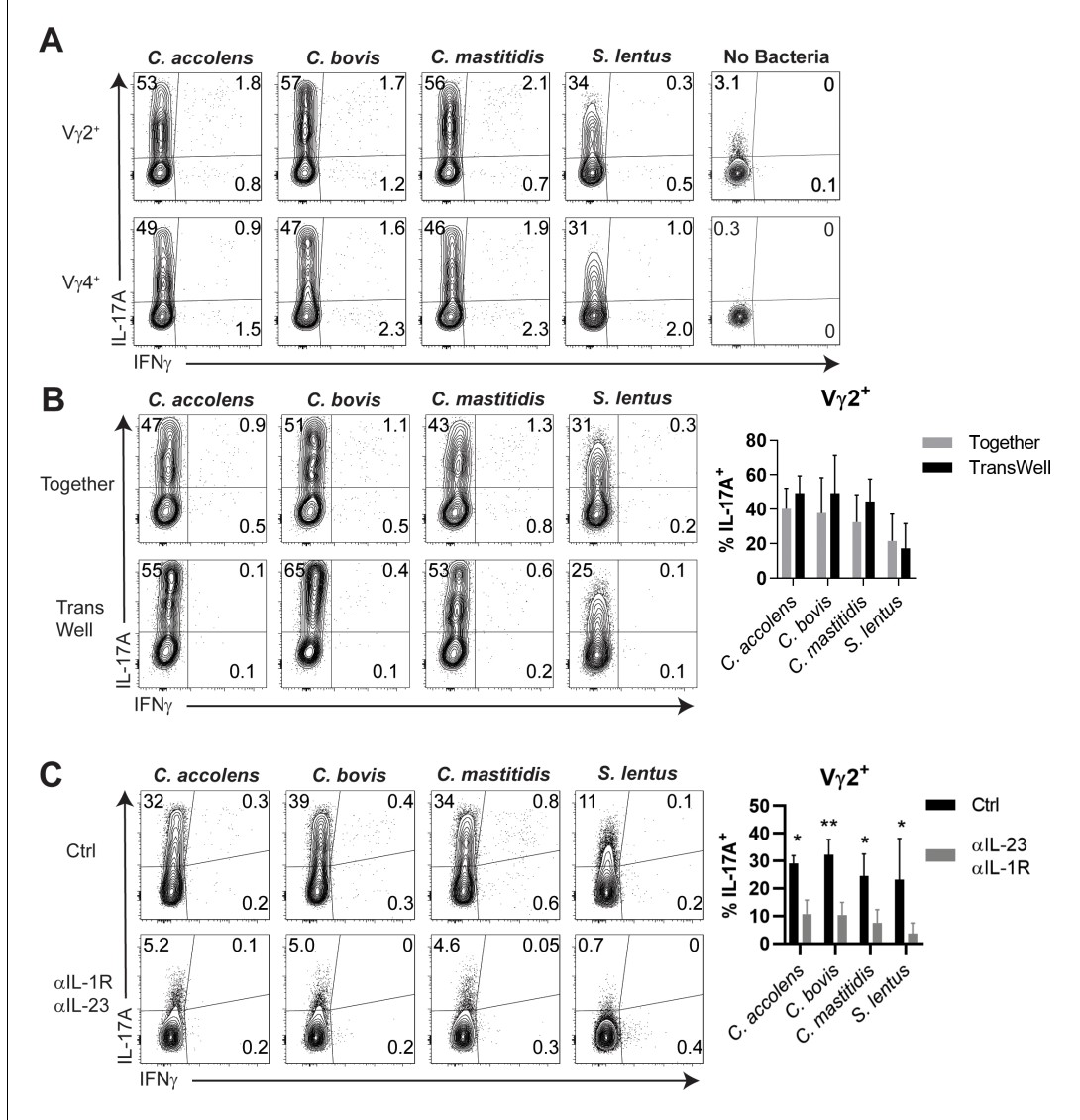

**Figure 5.** Cytokine-dependent, contact-independent Tγδ17 responses to skin commensals. (**A**) Total γδ cells were enriched from skin-draining LN and co-cultured with antigen presenting cells and the indicated heat-killed commensal bacteria at a 1:1:10 ratio for 16–18 hr, and then cultured for an additional 4 hr in the presence of Golgi Stop and Plug. IL-17A and IFNγ production was assessed by intracellular cytokine staining. Data are representative of 2 independent experiments. (**B**) Total γδ cells, splenic DC, and the indicated commensal bacteria were cultured at a 1:1:10 ratio as in (**A**) Together in a well or in a 0.4 micron TransWell apparatus in which DC and bacteria were placed in the top chamber and γδ cells were placed in the bottom chamber. Summary data are pooled from 2 independent experiments. (**C**) Cultures as above with 10 ug/mL each of anti-IL-1R and anti-IL-23 neutralizing Abs or isotype control Abs. Intracellular production of IL-17A and IFNγ was then assessed by FACS. Summary data are pooled from 4 independent experiments. *, p<0.05; **, p<0.01 by ANOVA.

Cytokine production from skin-infiltrating CD4[+] T cells was assessed to correlate effector function with the TCR Vβ repertoire. In WT mice, Th17 cells (IL-17[+] and IL-17/22[+]) were found in both Vβ4[+] and Vβ4[-] dermal CD4[+] T cell populations, but there was a biased representation of these effectors within Vβ4[+] T cells (*Figure 6D,E*).~10% of WT skin CD4[+] T cells were geared for IL-13 and/or IL-4 production, but there were negligible numbers of skin Th1 and Th22 cells (data not shown). In contrast, *Sox13[-/-]* AD skin lesions were enriched in Th17 subset and a larger population of dual IL-17[+]/ 22[+] Th17 cells, which were strongly biased to Vβ4[+] T cells (*Figure 6D,E*). Moreover, another AD-associated Th subset was the IL-22-only Th22 cells (*Czarnowicki et al., 2015*; *Fujita, 2013*), which predominantly expressed Vβ4 (*Figure 6E*). Frequencies and TCR Vβ repertoire of skin Th2 cells

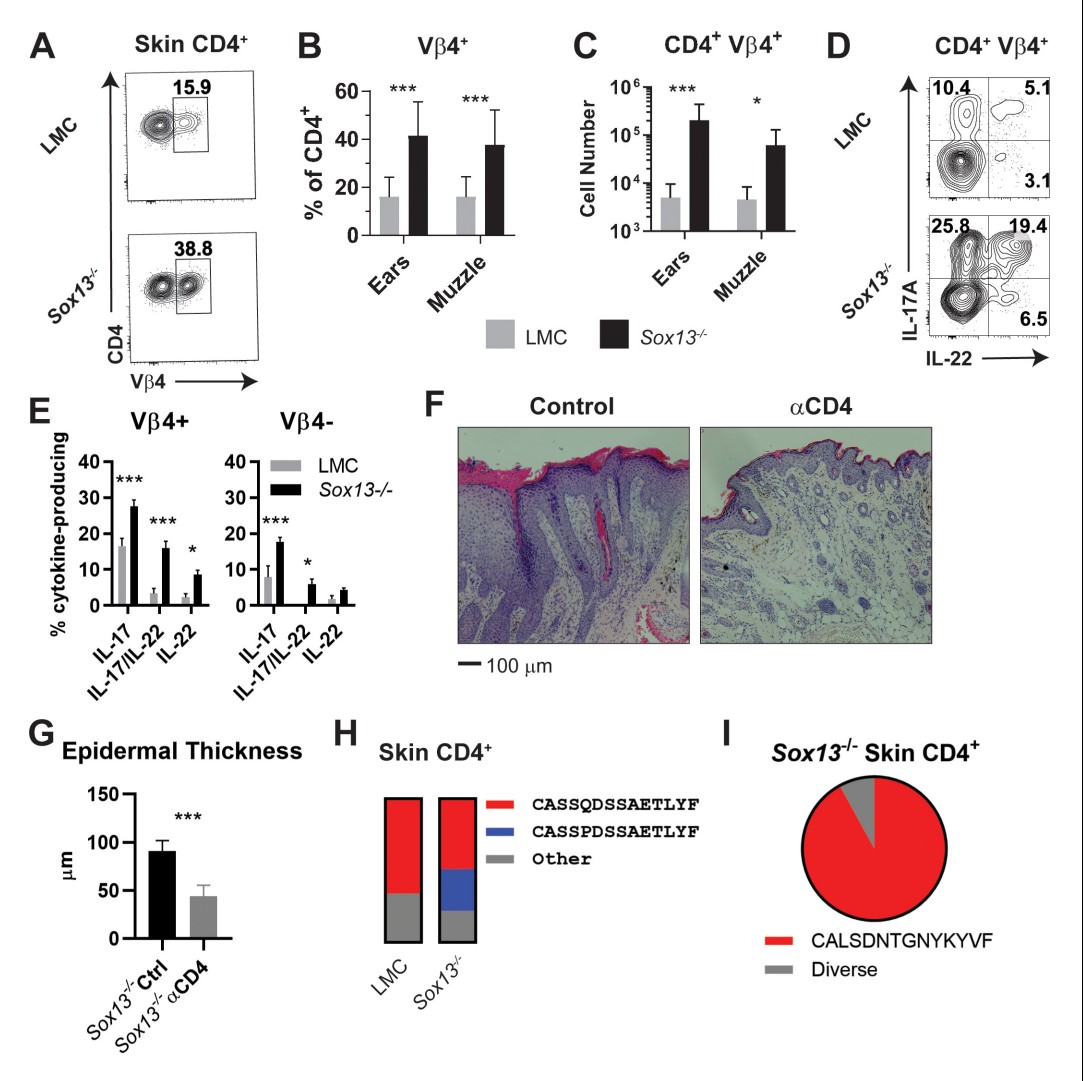

**Figure 6.** Expansion of dominant CD4$^+$ clonotypes in *Sox13$^{-/-}$* skin. (**A**) Muzzle-infiltrating cells were isolated from 5-6mo LMC and *Sox13$^{-/-}$* mice and analyzed for Vβ usage by CD4$^+$ T cells. (**B**) Summary data of Vβ4$^+$ frequency among skin-infiltrating CD4$^+$ cells in LMC and *Sox13$^{-/-}$* mice. n = 13–17 mice. ***, p<0.001 by ANOVA. (**C**) Enumeration of CD4$^+$ Vβ4$^+$ cells in LMC and *Sox13$^{-/-}$* skin. n = 6/group. ***, p<0.001; *, p<0.05 by ANOVA. (**D**) Skin-infiltrating cells were isolated from 5 mo mice, restimulated in vitro with PdBu/ionomycin, and IL-17 and IL-22 production by Vβ4$^+$ and Vβ4$^-$ CD4$^+$ T cells assessed via FACS. Data are representative of >4 experiments analyzing 2–3 mice/genotype/experiment. (**E**) Summary data of multiple experiments performed as in Panel D. n = 5–6 pooled from 3 independent experiments. ***, p<0.001; *, p<0.05 by ANOVA. (**F**) Starting at 3 months of age, *Sox13$^{-/-}$* mice were treated with control Ab (Ctrl) or a cell depleting Ab targeting CD4 antigen (αCD4) until 6 mo. AD disease severity was then assessed by H and E staining of muzzle skin. Data are representative of 10 mice treated with αCD4 Ab across 2 independent experiments. (**G**) Epidermal thickness in Ctrl and αCD4 Ab treated *Sox13$^{-/-}$* mice as assessed by analysis of histology images. n = 5 mice/group. ***, p<0.001 by Student's *t*-test. (**H**) Summary stacked bar charts of TCR Vβ4 CDR3 clonotype analysis of skin (ear and muzzle combined) infiltrating CD4$^+$ T cells in LMC and *Sox13$^{-/-}$* mice by deep sequencing, focusing on the two major clonotypes. Minimal 1 million reads/sample. Each stack reports proportion of each class on the right amongst total Vβ4 CDR3 sequence reads. (**I**) Summary of TCR Vα4 CDR3 clonotype analysis by pie chart of skin-infiltrating CD4$^+$ T cells in *Sox13$^{-/-}$* mice. LMC control not shown as there were insufficient reads.

The online version of this article includes the following source data and figure supplement(s) for figure 6:

**Source data 1.** TCR Vβ4 (TRBV2 by IMGT nomenclature) CDR3 sequencing analysis of CD4$^+$ non-Treg cells from the skin of LMC and *Sox13$^{-/-}$* mice as summarized in *Figure 6H*.

**Source data 2.** TCR Vα4 (TRAV6 by IMGT nomenclature) CDR3 sequencing analysis of CD4$^+$ non-Treg from the skin of *Sox13$^{-/-}$* mice as summarized in *Figure 6I*.

*Figure 6 continued on next page*

*Figure 6 continued*

**Figure supplement 1.** Expansion of CD4$^+$Vβ4$^+$ T cells in the skin of *Sox13$^{-/-}$* mice.

(~10% of CD4$^+$ T cells) and cytokine producing skin CD8$^+$ T cells (from WT and *Sox13$^{-/-}$* mice) were unchanged in *Sox13$^{-/-}$* mice at 3 and 6 mo (data not shown). To demonstrate that the expanded CD4$^+$ T cells critically contribute to AD, *Sox13$^{-/-}$* mice were treated with CD4 T cell depleting Abs starting at 3 mos of age for three mos. Skin inflammation significantly improved, including substantially reduced epidermal hyperplasia (*Figure 6F,G*) and amelioration of eosinophil and neutrophil infiltration (data not shown). Collectively, these results indicate that CD4$^+$ αβ T cells are the major driver of AD in *Sox13$^{-/-}$* mice and Vβ4$^+$ CD4$^+$ T cell expansion with enhanced IL-22 production is the primary distinguishing feature of αβ T cells in AD, dovetailing with findings in human severe AD (*Czarnowicki et al., 2015*).

To test the possibility of clonal TCRVβ4$^+$ T cell expansion, we used high throughput sequencing to identify TCR Vβ clonotypes expressed on conventional αβ T cells of WT and *Sox13$^{-/-}$* mice at 5 months of age. We interrogated cells expressing Vβ4 TCRs, as well as ones expressing Vβ2, 6 and 8. Collectively, these T cells represent 50–70% of CD44$^+$CD4$^+$ T cell repertoires. Analyses of skin from healthy mice revealed a single, dominant clonotype (CDR3β: CASS**Q**DSSAETLYF) expressed on ~70% of all CD44$^+$Vβ4$^+$CD4$^+$ T cells (*Figure 6H*). Notably, CD4$^+$ T cells expressing this TCR Vβ clonotype along with a related Vβ4 sequence (CDR3β: CASS**P**DSSAETLYF) were strongly expanded in diseased *Sox13$^{-/-}$* mice, making up >75% of Vβ4$^+$ conventional CD4+ T cells. These clonotypes, which we denote as the common Vβ4 (comVβ4), were less frequent in activated/memory T cells in dLNs (2–3% of CD4+ T cells), and detectable only at minute frequencies in naïve T cells (<0.1%). In comparison, TCRβ chains expressed on skin-resident CD8$^+$ T cells in WT and *Sox13$^{-/-}$* mice were oligoclonal (data not shown).

To begin to identify TCRαβ clonotypes, 15 CD4$^+$ skin T cell lines were established from *Sox13$^{-/-}$* AD skin and converted to hybridomas. Although this approach was inefficient, four Vβ4$^+$ T hybridomas expressed the comVβ4 chain, all of which were paired with a conserved Vα4.9 chain (comVα4, CALSDNTGNYKYVF). TCRα deep sequencing of total skin CD4$^+$ T cells confirmed that >90% of the Vα4+ cells expressed the comVα4 chain in the skin of diseased *Sox13$^{-/-}$* mice (*Figure 6I*), while these clonotypes were rare in dLNs. Together, these studies reveal that ~25% of all skin CD4$^+$ T cells in AD mice express two related TCR clonotypes composed of Vβ4Vα4.9 TCR, indicative of antigen-specific clonal expansion.

## Discussion

When there are systemic structural breakdowns of the skin barrier, dysregulated immunity leads to uncontrolled inflammation. Most mouse models of AD to date involve either a systemic breakdown of the skin barrier (e.g. filaggrin/matted [*Kawasaki et al., 2012*; *Saunders et al., 2013*] or *Adam17*-deficient [*Kobayashi et al., 2015*]), or rely on heavy manipulations of the skin (e.g. tape stripping followed by antigen challenge (*Oyoshi et al., 2012*) or topical applications of inflammatory cytokines, such as IL-23 [*Li et al., 2016*]) and they may not reflect natural progression of AD. In particular, physiological events that contribute to skin barrier damage in postnatal animals have not been modeled for experimentation. Moreover, identification of pathogenic CD4 T cell clones and events that trigger adaptive T cells that culminate in AD have not yet been systematically investigated. Here, removal of one innate dermal T cell sentinel subset that normally populates the neonatal skin is sufficient to cause spontaneous, highly penetrant AD, with many of the major hallmarks of the human disease. Early changes in basal keratinocyte transcriptome, well before the onset of fulminant disease, are consistent with an altered barrier formation that is likely to have linkage CB dysbiosis and the damaging immune responses that ensue. Our model thus serves to close fundamental gaps in understanding of AD and identify dermal innate Vγ2$^+$ Tγδ17 cells as skin regulatory T cells.

AD in *Sox13$^{-/-}$* mice is driven by Th17 cells, and transfer studies using CD44$^{hi}$ T cells from dLNs of diseased *Sox13$^{-/-}$* mice did result in AD-like symptoms in *Sox13$^{-/-}$*, but not in LMC host (data not shown). However, the hosts need to be primed for the disease transfer, by sublethal irradiation and skin scarring, and the kinetics of disease induction and severity were variable. Predictable kinetics of

AD transfer using dermal T cells from diseased *Sox13*⁻/⁻ mice would be ideal, and we are currently attempting to establish more physiological priming conditions for these studies. An important implication of our findings is that homeostasis-maintaining Tγδ17 cells cannot be substituted by other type 3 cytokine-polarized T cells: loss of Vγ2⁺ Tγδ17 cells leads to expansion of Th17-like αβ T cells that are associated with inflammation. Accordingly, further dissection of the pathways that are selectively engaged under homeostasis versus inflammation will be important to determine whether homeostasis-specific features could be harnessed to enhance resolution of inflammation. γδ T cells as a population have been suggested to regulate inflammation and αβ T cell responses in mucosal tissues, based on dysregulated type 2 cytokine responses in the lung of *Tcrd*⁻/⁻ mice (*Zuany-Amorim et al., 1998*; *Lahn et al., 1999*; *Guo et al., 2018*), spontaneous skin inflammation in *Tcrd*⁻/⁻: FVB background (*Girardi et al., 2002*), and keratitis in *Tcrd*⁻/⁻:B10 mice (*O'Brien et al., 2009*). However, γδ T cell subset-specific function in these disease models were unknown and Tγδ17 cells have been considered principally as an inflammatory/pathogenic cell type, required for psoriasis and EAE (*Sutton et al., 2009*), ocular responses to *C. mastitidis* to protect against fungal infection (*St Leger et al., 2017*), and intestinal responses to *Listeria* (*Sheridan et al., 2013*). In the frontline mucosal tissues there are several innate and conventional T cells that can produce IL-17, including Tγδ17, MAITs, iNKT17, ILC3, Tc17 and Th17 cells. While the homeostatic role of IL-17 in the skin was documented (*Qian et al., 2004*; *Matsushima et al., 2010*; *Floudas et al., 2017*) the critical cell source of IL-17 was not known. We show that Vγ2TCR⁺ Tγδ17 cells is that source necessary to prevent the skin CB dysbiosis dependent inflammation cascade.

Given that there exists multiple innate type 3 cytokine producing T cells, it remains unclear why Vγ2⁺ Tγδ17 cells are indispensable in skin homeostasis. The alternate fetal-derived PLZF⁺ Vγ4TCR⁺ Tγδ17 cells in adipose tissues are required for normal thermogenic responses (*Kohlgruber et al., 2018*) and they are also present in most mucosal tissues (*Jin et al., 2019*). While *Sox13*⁻/⁻ mice generate reduced numbers of Vγ4⁺ Tγδ17 cells from the postnatal thymus, their numbers in peripheral tissues normalize over time (*Malhotra et al., 2013*). Thus, in both AD and psoriasis models, Vγ2⁺ Tγδ17 cells are the critical mediators of skin homeostasis and acute inflammation, and the Vγ4⁺ counterpart, and other innate IL-17 producing T cells such as DN MAITs that are present in the dermis of *Sox13*⁻/⁻ mice, cannot functionally compensate for the loss of Vγ2⁺ Tγδ17 cells. So far, only two molecular features distinguish these two γδ T cell subtypes in the dermis: highly biased expression of the scavenger receptor Scart2 on Vγ2⁺ Tγδ17 cells (*Narayan et al., 2012*) and the distinct TCRs. The nature of ligands recognized by these receptors is unknown, but given the independence of Vγ2⁺ Tγδ17 cells from skin CB for their development and persistence in the skin, the likelihood of novel environmental cues determining their function is high.

We are actively investigating the ligand recognized by Vβ4/Vα4 clonotype T cells. Skin Th17 cells are absent in GF mice, and their numbers are restored to the normal range in Abx *Sox13*⁻/⁻ mice, indicating their dependence on CB. However, skin T cell hybridomas expressing the clonotypic Vβ4 TCR did not respond to various *Staphyloccus* and *Corynebateria* species, suggesting these cells respond to other CB or skin antigens. Skin CD8⁺ Tc17 cells recognize *S. epidermidis*-derived *N*-formyl methionine peptides presented by the non-classical MHC-Ib molecule H2-M3 (*Linehan et al., 2018*) and it is possible that Vβ4/Vα4 clonotypes also recognize non-conventional MHC molecules. MR1-5-OP-RU or CD1d-PBS tetramers did not stain skin Vβ4⁺ T cells, ruling out the obvious candidates as likely ligands.

The emerging model of AD progression is then that tonic IL-17/22 produced by Tγδ17 cell recognitions of CB and other skin-specific cues promote normal development of keratinocytes in postnatal mice. In the absence of this lymphoid-epithelial crosstalk, skin CB dysbiosis develops in conjunction with altered skin barrier, driving APC activation and setting in motion aggressive activation, infiltration and expansion of type 3 cytokine producing T cells that are primarily focused on dealing with altered CB, but also result in collateral skin degradation. In parallel, damaged skin releases DAMPs, such as IL-33 that activates ILC2, which in turn promote Th2 responses (*Salimi et al., 2013*). Cytokines and chemokines copiously produced by activated skin lymphocytes perpetuate eosinophilia and neutrophilia that chronically worsen skin damage. In this setting, skin Tregs do not significantly impact the disease progression, as their sustained depletion does not impact AD amelioration caused by conventional CD4 T cell ablation (*Figure 4*).

Human inflammatory skin diseases also involve Tγδ17 cells (*Laggner et al., 2011*). While type 3 cytokine producing lymphocytes have been implicated in human AD progression and maintenance,

the role of early IL-17 in human neonates in skin barrier maintenance has not been investigated. Emerging evidence for preprogrammed effector T cells in the gut and blood at the fetal stage (*Zhang et al., 2014*; *Schreurs et al., 2019*; *Li et al., 2019*) support the possibility that human and rodents share the similar lymphoid lineage developmental blueprint to generate pre-programmed lymphoid effectors early in life. Whether Tγδ17 cells are the main producers of IL-17 in human skin requires definitive resolution, but a priori, any type 3 cytokine producing innate lymphocytes in the skin, including MAITs, iNKT17 and ILC3 can serve the regulatory function of murine dermal Tγδ17 cells, likely to be dependent on the commensal community as well as genetic and environmental variations in skin fitness in the outbred populations. Clinically, IL-17 blockade is being tested to treat skin inflammatory disorders. Emerging findings of regulatory function of IL-17 in the skin raise the possibility of potential negative impacts on skin barrier function, aggravated by the emergence of IL-22 as the pathogenic effector in fulminant skin inflammatory disorders with interference of IL-17R signaling.

## Materials and methods

### Mice

All mice were housed in specific pathogen-free (SPF) conditions, and all procedures were approved by the University of Massachusetts Medical School (UMMS) IACUC. *Sox13*$^{-/-}$ mice have been described previously (*Melichar et al., 2007*), and are maintained on a 129S1/SvlmJ (129) background as C57BL/6 (B6).*Sox13*$^{-/-}$ mice are embryonic lethal. B6, B6.129P2-*Tcrb*$^{tm1Mom}$/J (B6.*Tcrb*$^{-/-}$), *Rora*$^{-/-}$ and B6.*Il17a*$^{tm1Bcgen}$ (B6.*Il17a-Egfp*) mice were purchased from Jackson Laboratories. Germ-free B6/129 mice were from HDDC Gnotobiotics Core, Harvard. To generate 129.*Sox13*$^{-/-}$*Tcrb*$^{-/-}$, B6. *Tcrb*$^{-/-}$ was backcrossed to 129 mice for 9 generations, and then intercrossed with *Sox13*$^{-/-}$ mice to generate double knockout mice. B6.Cg-Thy1a/Cy Tg(*Tcrab*)8Rest/J (PMEL Tg) mice were kindly provided by John Harris (UMMS).

### Cell isolation and stimulation and antibodies

Ears and muzzle skin were first treated with Nair for 2 min, and then Nair was gently wiped away with a PBS-moistened cotton-tip applicator, and tissue was subsequently rinsed extensively with PBS prior to digestion. For this study, muzzle tissue is demarcated by the boundaries of the vibrissiae. Ears were split into dorsal and ventral halves, and muzzle tissue was removed of subcutaneous tissue. Skin was finely minced and then digested with 1 U/mL Liberase TL (Roche) + 0.5 mg/mL Hyaluronidase (Sigma-Aldrich) + 0.05 mg/mL DNAse (Roche) dissolved in HBSS (with $Ca^{2+}$/$Mg^{2+}$, Corning) + 10 mM HEPES (Gibco) + 5% FBS (Sigma-Aldrich) for 90 min at 37°C with gentle shaking. After digestion, EDTA (Sigma-Aldrich) was added at 5–10 mM, and then tissue was strained through a 100 μm cell strainer. Cell were washed in FACS buffer (DPBS, $Ca^{2+}$/$Mg^{2+}$-free + 0.5% BSA [Fisher Scientific] + 2 mM EDTA) and then plated for antibody staining. Mandibular and parotid dLN were mechanically homogenized between etched glass slides (Fisher Scientific) and strained through 70 μm mesh prior to plating for antibody staining.

The following anti-mouse antibodies were purchased from Biolegend, BD Biosciences, or Thermo-Fisher and used for FACS analysis: CD45 (30-F11), Siglec F (S17007L), Ly-6G (1A8), Ly-6C (HK1.4), MHC II (M5/114.15.2), CD3 (17A2), CD5 (53–7.3), B220 (RA3-6B2), CD11b (M1/70), Gr-1 (RB6-8C5), Ter-119 (Ter-119), Thy1.2 (30-H12), F4/8) (BM8), TCRβ (H57-597), CD4 (GK1.5), CD8β (YTS156.7.7), PD-1 (29F.1A12), CXCR5 (2G8), GL7 (GL7), CD95 (Jo2), CD138 (281-2), IgD (11–26 c.2a), Vβ4 (KT4), CD49f (GoH3), TCRδ (GL3), Vγ2 (UC3-10A6), Vγ3 (536), CCR6 (140706), IL-17A (17B7), IL-22 (poly5164), IFNγ (XMG1.2), IL-4 (11B11), IL-5 (DIH37), IL-13 (eBio13A), FoxP3 (FJK-16s), GATA3 (TWAJ), RORγt (AFKJS-9), Bcl6 (K112-91), Ki-67 (B56). MR1 and CD1d tetramers were provided by the NIH Tetramer Core Facility at Emory University. All samples were labeled with a fixable viability dye (ThermoFisher) prior to analysis. The combinatorial TCR Vβ staining strategy has been described previously (*Diz et al., 2012*), and all Vβ epitopes were found to be resistant to the enzymes used for digestion when tested on dLN cells (data not shown). CellTrace Violet was purchased from Thermo-Fisher, and cell were labeled as recommended by the manufacturer. For intracellular cytokine staining, cells were fixed/permeabilized with Cytofix/Cytoperm buffer (BD Biosciences) and then stained

**Table 1.** PCR Primers used in this study.

| Sequence | F/R | Description |
|---|---|---|
| CCTGGACTCTCCACCGCAA | F | Il17a |
| TTCCCTCCGCATTGACACAG | R | Il17a |
| TTTCCTGTCTGTATTGAGAAACCT | F | Il33 |
| TATTTTGCAAGGCGGGACCA | R | Il33 |
| CGCTTGAGTCGGCAAAGAAAT | F | Il1a |
| TGGCAGAACTGTAGTCTTCGT | R | Il1a |
| GCCACCTTTTGACAGTGATGAG | F | Il1b |
| GACAGCCCAGGTCAAAGGTT | R | Il1b |
| TCCTCTCTGCAAGAGACTTCC | F | Il6 |
| TTGTGAAGTAGGGAAGGCCG | R | Il6 |
| AGCTGTAGTTTTTGTCACCAAGC | F | Ccl2 |
| GTGCTGAAGACCTTAGGGCA | R | Ccl2 |
| TCACAGCAACGAAGAACACCA | F | Il4 |
| CAGGCATCGAAAAGCCCGAA | R | Il4 |
| CAAGCAATGAGACGATGAGGC | F | Il5 |
| GCATTTCCACAGTACCCCCA | R | Il5 |
| CACTACGGTCTCCAGCCTCC | F | Il13 |
| CCAGGGATGGTCTCTCCTCA | R | Il13 |
| CACCAGCGGGACATATGAATCT | F | Il23a |
| CACTGGATACGGGGCACATT | R | Il23a |
| TTGAGGTGTCCAACTTCCAGCA | F | Il22 |
| AGCCGGACGTCTGTGTTGTTA | R | Il22 |
| AGAGTTTGATCCTGGCTCAG | F | 16S V1 Universal Primer 27F |
| ATTACCGCGGCTGCTGG | R | 16S V3 Universal Primer 534R |
| AAGCCTGATGACTCGGCCACA | F | Vb4 TCR deep seq |
| CTTGGGTGGAGTCACATTTCTCAGATCCTC | R | Cbeta TCR deep seq |
| AACTGTACTTATTCAACCACA | F | Va4 TCR deep seq |
| CTGTGAACTGTTCCTATGAAACC | F | Va4 TCR deep seq |
| TAAACTGTACTTATTCAACCACA | F | Va4 TCR deep seq |
| CCTGATAATAAATTGCACGTATTCA | F | Va4 TCR deep seq |
| GGTACACAGCAGGTTCTGGGTTCTGGATG | R | Calpha TCR deep seq |

in permeabilization buffer. For intranuclear transcription factor staining, cells were fixed/permeabilized and then stained using the FoxP3/transcription factor Staining Buffer Set (eBioscience).

For in vitro restimulation, digested skin cells or dLN cells were resuspended in complete DMEM-10 medium (DMEM, high glucose + 10 mM HEPES + 4 mM L-Glutamine + 1x non-essential amino acids + 1 mM sodium pyruvate + 100 U/mL penicillin + 100 ug/mL streptomycin (all Gibco) + 10% FBS) and cultured with 500 ng/mL phorbol 12,13 dibutyrate (PdBu, Tocris) + 1 μM Ionomycin (Sigma-Aldrich) + 1x GolgiStop + 1x GolgiPlug (BD Biosciences) for 2–3 hr at 37℃. After stimulation, cells were washed in FACS buffer and then stained with antibodies as indicated above. Serum IgE was assessed by ELISA (BioLegend). To deplete CD4[+] T cells, mice were injected i.p. with anti-CD4 (GK1.5, Bio X cell) or rat IgG2b isotype control. Initially, mice received two doses of 500 μg/mouse on day 0 and day 2. Thereafter, mice received a weekly maintenance dose of 100 μg/mouse to maintain depletion. Depletion was confirmed by analysis of dLN and skin T cells stained with anti-CD4 clone RM4-4 (Biolegend), which binds a non-overlapping epitope.

## Histology and immunofluorescence microscopy

For H and E staining, muzzle tissue was first fixed in 10% neutral-buffered formalin for 24 hr, and then paraffin embedded, sectioned, and stained by the UMMS DERC Morphology Core. Epidermal thickness was calculated using ImageJ, taking the average of 3 measurements per image to record as 1 data point. For immunofluorescence microscopy, dLN were fixed in 4% paraformaldehyde (diluted from 16% ampules, Electron Microscopy Sciences) in PBS for 6–8 hr at 4°C, washed three times in PBS, equilibrated in 30% sucrose in PBS overnight, and then frozen in OCT compound (Sakura Tissue-Tek). Cryosections were cut to 7 um thickness, blocked in PBS + 0.3% Triton X-100 + 5% normal mouse serum for 1 hr at RT, then endogenous biotin was blocked using the Avidin/Biotin Blocking System (BioLegend) as recommended. Primary antibody labeling was performed in blocking buffer overnight at 4°C in a humidified chamber using the following antibodies: anti-CD4 Alexa Fluor 647 (BioLegend), goat anti-IgD purified (Cedarlane Labs), anti-GL7 Alexa Fluor 488 (BioLegend), and anti-CD11c Brilliant Violet 421 (BioLegend). Slides were washed 3x in PBS, and then labeled with donkey anti-goat Cy3 (Jackson ImmunoResearch) in blocking buffer for 1 hr at RT. Slides were rinsed 3x in PBS and mounted using Fluoromount-G (Southern Biotech). Images were acquired on a Zeiss Axio Observer with LED excitation using ZEN software (Zeiss) and displayed using best-fit parameters.

## TCR CDR3 deep sequencing

The strategy for deep sequencing of TCR Vβ4 CDR3 regions has been described previously (*Stadinski et al., 2016*). Cells from pooled muzzle and ear skin of 6 mo LMC and $Sox13^{-/-}$ mice with AD were sorted via FACS as Live CD45$^+$ TCRβ$^+$ CD4$^+$ CD25$^-$ GITR$^{lo}$ to exclude Treg cells. RNA was extracted using Trizol (ThermoFisher), and cDNA generated using oligo dT priming and OminScript reverse transcriptase (Qiagen) per the manufacturers' recommendations. PCR was performed using a Vβ4- or Vα4-specific forward primer containing adapter and barcode sequences combined with a Cβ or Cα reverse primer. Multiple forward primers were used for Vα4 to ensure coverage of the entire Vα4 family. Sequencing was performed on an Illumina MiSeq at the Deep Sequencing Core Lab. For analysis, low quality (Q score <25) reads were removed and then sequences were parsed based on the sample barcode using fastq-multx. TCR V and J nucleotide sequences were converted to amino acid sequences using TCRKlass, using the conserved Cys residue of TCR Vβ to identify CDR3 position 1.

## Microbiome sequencing, antibiotics, and in vitro bacterial/γδ cell cultures

To sequence the muzzle microbiome of LMC and $Sox13^{-/-}$ mice, sterile cotton-tip applicators were swabbed across both sides of the muzzle and then placed into sterile Eppendorf tubes and placed onto dry ice. Muzzle swabs were sent to Molecular Research LP (MR DNA, Shallowater, TX) for DNA extraction and sequencing on an Illumina MiSeq. Extracted DNA was used to amplify the 16S V4 region, and then amplicons were purified for library generation. For analysis, low quality and short sequences (<150 bp) were removed. Operational taxonomic units were identified and classified using BLASTn and a curated database derived from NCBI, RDPII, and GreenGenes. Count files were then converted to percentages by dividing the number of counts for a given phylum/species by the sum of all counts. For antibiotic treatment, $Sox13^{-/-}$ breeders were placed on drinking water containing 0.5 mg/mL enrofloxacin and 0.5 mg/mL cefazolin (hereafter Abx). Weaned mice were then placed on Abx water and analyzed at six mo. To assess γδ cell responses to skin commensals, LN γδ T cells were isolated from WT 129 mice by negative selection (without the use of anti-TCRδ Abs). CD11c$^+$ cells were isolated from spleens using CD11c microbeads (Miltenyi Biotec). *Corynebacteria* were grown on brain heart infusion agar (BHI) with 1% Tween-80, then grown in BHI broth with 1% Tween-80 overnight. *Staphylococcus* was grown on trypticase soy agar, then grown in BHI broth overnight. *C. accolens* was purchased from ATCC. *C. bovis* and *C. mastitidis* were kindly provided by K. Nagao (National Institute of Arthritis and Musculoskeletal and Skin Diseases, 9). *S. lentus* was isolated from the muzzle skin of a $Sox13^{-/-}$ mouse with AD by streaking onto mannitol salt agar, followed by re-streaking of an isolated, mannitol-fermenting colony. Species identification was determined by sequencing analysis of 16S V1-V3 followed by BLAST. The day of the experiment, bacterial cultures were subcultured 1:100 for 2–4 hr to permit recovery into exponential growth phase.

Culture density was determined by OD600, and then bacteria were resuspended in PBS and heat-killed at 56°C for 1 hr. DC, γδ T cells, and bacteria were cultured at 1:1:10 ratio for 16–18 hr, and then GolgiStop and GolgiPlug were added for an additional 4 hr prior to FACS analysis. In some cases, anti-IL-23 (MMp19B2, BioLegend) and anti-IL-1R (JAMA-147, Bio X Cell) or isotype control antibodies were added for the entire culture duration. To assess contact dependency, DC and bacteria were placed in the top chamber of a 0.4 μm TransWell apparatus (Corning) and γδ T cells in the bottom well.

## Gene expression analysis

For RT-qPCR analysis of whole skin, skin was excised and stored in RNALater (ThermoFisher) overnight at 4°C. The next day, the sample was homogenized in Trizol using an Omni Tissue homogenizer, and then RNA isolated. RNA was converted to cDNA using oligo dT priming and AffinityScript reverse transcriptase (Agilent). qPCR was performed using iQ SYBR green Supermix and a CFX96 thermal cycler (Bio-rad), followed by thermal melt curve analysis to confirm specific amplification. Primers used in this study were synthesized by Integrated DNA Technologies and are reported in *Table 1*. For RNA sequencing analysis, epidermal keratinocytes were purified by first separating dorsal and ventral halves of dissected ears and floating dermis down on 5 U/mL dispase (Sigma-Aldrich) with 0.05 mg/mL DNAse I for 50 min at 37°C. Epidermis was then peeled away, and the dermis discarded. The Epidermis was further minced and then digested for an additional 30 min with 2 mg/mL Collagenase IV (Worthington) with 0.05 mg/mL DNAse I. Epidermal single cell suspensions were then labeled with anti-CD49f to identify basal keratinocytes, anti-CD45 to exclude leukocytes, and 7-AAD to exclude dead cells. Keratinocytes were double-sorted for purity, with the second sort into cell lysis buffer for RNA extraction at $10^4$ cell equivalents. Samples were generated in triplicates. RNAseq analyses were performed by the Immunological Genome Project, using the standard operating protocol (Immgen.org). Volcano plots and DEG lists were generated using MultiPlot Studio (part of the GenePattern from the Broad Institute). Gene Ontology (GO) terms were identified using the DAVID bioinformatics resource (https://david.ncifcrf.gov/), with significance determined by EASE score (a modified Fisher Exact).

## Statistical analysis

Graphing and statistical analysis was performed using GraphPad Prism software. Significance values, tests used, and cohort sizes are indicated in figure legends. Unless otherwise indicated, comparison of two groups was analyzed by unpaired two-tailed Student's *t* test, and comparison of three or more groups was analyzed by ANOVA with Sidak's correction for multiple hypothesis testing.

## Acknowledgements

We thank J Harris (UMMS) and K Nagao (NIH) for reagents, the Immunological Genome Project Consortium for RNAseq data generation, A Reboldi for critical reading of the manuscript, and UMMS FACS Core for cell sorting. Supported by NIH grants AI101301, AR071269 to JK.

## Additional information

### Funding

| Funder | Grant reference number | Author |
|---|---|---|
| National Institutes of Health | AI101301 | Joonsoo Kang |
| National Institutes of Health | AR071269 | Joonsoo Kang |

The funders had no role in study design, data collection and interpretation, or the decision to submit the work for publication.

### Author contributions

Nicholas A Spidale, Conceptualization, Data curation, Formal analysis, Investigation, Visualization, Methodology, Writing - original draft, Writing - review and editing; Nidhi Malhotra, Bing Miu, Eric

Huseby, Formal analysis, Investigation; Michela Frascoli, Formal analysis, Visualization; Katelyn Sylvia, Data curation, Formal analysis; Coral Freeman, Data curation, Visualization; Brian D Stadinski, Data curation, Methodology; Joonsoo Kang, Conceptualization, Data curation, Supervision, Funding acquisition, Writing - original draft, Project administration, Writing - review and editing

### Author ORCIDs
Nicholas A Spidale (iD) https://orcid.org/0000-0002-1568-4746
Joonsoo Kang (iD) https://orcid.org/0000-0001-8419-7995

### Ethics
Animal experimentation: This study was performed in strict accordance with the recommendations in the Guide for the Care and Use of Laboratory Animals of the National Institutes of Health. All experiments performed were approved by the University of Massachusetts Medical School IACUC (Protocol A1206).

### Decision letter and Author response
Decision letter https://doi.org/10.7554/eLife.51188.sa1
Author response https://doi.org/10.7554/eLife.51188.sa2

## Additional files

### Supplementary files
• Transparent reporting form

### Data availability
All data generated or analysed during this study are included in the manuscript and supporting files. Source Data files are provided for keratinocyte RNA-seq analysis, TCR sequencing, and skin microbiome analysis.

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
