## [Decision Letter]

**Acceptance summary:**

This study used genetically modified mouse models to demonstrate that a tissue resident IL-17 producing γδ T cell subset (nTγδ17) plays a crucial role in maintaining skin microbiota and barrier integrity. The findings of this study are significant as they provide mechanistic insights into the crosstalk between nTγδ17 cells and keratinocytes and reveal a potential role of nTγδ17 cells in regulating atopic dermatitis.

**Decision letter after peer review:**

Thank you for submitting your article "Neonatal-derived IL-17 producing dermal γδ T cells are required to prevent spontaneous atopic dermatitis" for consideration by *eLife*. Your article has been reviewed by three peer reviewers, one of whom is a member of our Board of Reviewing Editors, and the evaluation has been overseen by Tadatsugu Taniguchi as the Senior Editor. The reviewers have opted to remain anonymous.

The reviewers have discussed the reviews with one another and the Reviewing Editor has drafted this decision to help you prepare a revised submission.

Summary:

The manuscript by Spidale et al. describe a Sox13 knockout mouse model, which lacks innate IL-17 producing γδ T cells (nTγδ17) in the dermis, and spontaneously develops an atopic dermatitis like skin inflammation. The disease was dependent upon αβ T cells and commensal bacteria, and increases in granulocytes, macrophages, and serum IgE were also noted. The deficiency in dermal nTγδ17 cells was accompanied by changes in gene expression among keratinocytes, and in altered skin commensal bacteria. In addition, *Sox13^-/-^* skin lesions were enriched in Th17 subset, which were strongly biased to Vβ4^+^ T cells and likely resulted from antigen-specific clonal expansion. Overall, the experiments are well performed. The findings that nTγδ17 cells are important in maintaining skin microbiota and the nTγδ17 cell/keratinocyte crosstalk is crucial for skin barrier integrity are quite novel and interesting. However, the manuscript often failed to include results of statistical tests showing significance. In addition, it is not clear whether skin inflammation observed in this mouse model represent human atopic dermatitis.

Essential revisions:

1) Many of the figures do not include data showing that the findings are statistically significant. While it is nice to see typical flow profiles, bar graphs should be added, showing averages obtained for each group. This include Figures 1B, 1G, 2B, 2C, 2G, 4B, 4D, 4E, 5E, 5C, and 6H, and figure supplements Figure 2—figure supplement 1A-E, Figure 4—figure supplement 1B, C, and Figure 6—figure supplement 1A. In addition, Figure 4—figure supplement 1D shows an important result – the dependence of the presence of dermal Vγ4^+^ cells on commensal bacteria – and perhaps this should appear instead in a regular figure.

2) The authors repeatedly refer to "innate dermal γδ T cells, " and include in this group both the Vγ4^+^ and Vγ2^+^ subsets. But it is somewhat controversial whether the dermal Vγ4^+^ cells are actually "innate," since Cai et al. (Nat. Commun. 2014) showed that they can be reconstituted from adult bone marrow. Later in paper, the authors distinguish these subsets as fTγδ17 (for the fetal derived Vγ4^+^ cells) and nTγδ17 (for neonatal Vγ2^+^ cells). This is awkward and confusing; why not just refer to them throughout as Vγ2^+^ Tγδ17 and Vγ4^+^ Tγδ17 instead?

3) The authors do not show that Vγ4^+^ cells remain in the dermis in Sox13^-/-^ mice, or show whether dermal Vγ4^+^ cells are actually expanded and producing IL17 in these mice. Staining to verify that dermal Vγ2^+^ cells are missing in the *Sox13^-/-^*mice should also be added.

4) Regarding the AD model, the authors claim a new spontaneous atopic dermatitis model with all hallmarks of human disease. To make this claim, there should have been a more extensive characterization of the skin disease, beyond H&E staining and quantification of leukocytic infiltrate. Hallmarks of human disease that are missing (or not described?): such as itch, increased TH2 cells (IL-4/-5/-13/-31), barrier dysfunction, and increased mast cells. In addition, it would be useful to show photo images of the dermatitis in *Sox13^-/-^*animals.

5) Since *Sox13^-/-^*mice in other backgrounds (B6) are embryonic lethal, suggesting that the animals have other developmental defects besides nTγδ17 T cell subset. The author should address whether there are other (developmental) defects in the *Sox13^-/-^*mice. In addition, the authors should comment on whether keratinocytes or keratinocyte progenitors express Sox13?

6) The authors should de-emphasize the potential involvement of the observed clonotype (within Vβ4^+^ cells) in disease, since it is also highly abundant in non-diseased controls (Figure 6H). In addition, do CD4^+^Vβ4^+^ T cells enriched in the skin of *Sox13^-/-^*mice produce IL-17 and/or IL-22 in response to skin commensals?

7) In Figure 2—figure supplement 1H, please clarify why IL-17 is decreased in the skin in *Sox13^-/-^*mice.

---

## [Author Response]

Essential revisions:1) Many of the figures do not include data showing that the findings are statistically significant. While it is nice to see typical flow profiles, bar graphs should be added, showing averages obtained for each group. This include Figures 1B, 1G, 2B, 2C, 2G, 4B, 4D, 4E, 5E, 5C, and 6H, and figure supplements Figure 2—figure supplement 1A-E, Figure 4—figure supplement 1B, C, and Figure 6—figure supplement 1A. In addition, Figure 4—figure supplement 1D shows an important result – the dependence of the presence of dermal Vγ4^+^ cells on commensal bacteria – and perhaps this should appear instead in a regular figure.

We have provided representative flow cytometry profiles where we felt the phenotypes were clear-cut, but we can appreciate the reviewers’ interest in more complete data sets. We have provided summary data with statistical analysis as requested as described below for specific Figure panels:

Figure 1B – was originally summarized as cell count data in Figure 1C-F.

Figure 1G – A summary panel added below flow cytometry data.

Figure 2B – A summary added as new Figure 2C.

Figure 2C – Summary graphs placed below flow cytometry plots (now Figure 2D).

Figure 2G, Figure 4B – For histology, we now show replicate results, rather than an arbitrary quantitations, which we are not convinced are informative.

Figure 4D – Summary added as new Figure 4E.

Figure 4E – Summary graph added under flow cytometry data (now Figure 4F).

Figure 5B – Summary graph added to the right of flow cytometry data.

Figure 5C – Summary graph added to the right of flow cytometry data.

Figure 6H –For TCRβ chain sequencing data, rather than more replicates we chose to use the T cell hybridoma approach, to directly examine true clonotypes (that is TCR α/β pairs, not just β chain abundance), as mentioned in the Discussion.

Figure 2—figure supplement 1A-E – All summaries added; Treg, GC, PC FACS FACS plots now replaced with summaries.

Figure 4—figure supplement 1 – A new summary of data provided.

Figure 2—figure supplement 1C – Requested summary was already provided.

Figure 6—figure supplement 1A – Requested summary was already provided (a heatmap to be succinct and visually simple) the only statistically significant difference (CD4 Vβ4) is denoted by asterisks.

Regarding Figure 2—figure supplement 1D, we chose to include these data in the supplemental section rather than as a main figure because similar findings have been reported previously by the Kasper lab (see references cited in the text).

2) The authors repeatedly refer to "innate dermal γδ T cells, " and include in this group both the Vγ4^+^ and Vγ2^+^ subsets. But it is somewhat controversial whether the dermal Vγ4^+^ cells are actually "innate," since Cai et al. (Nat. Commun. 2014) showed that they can be reconstituted from adult bone marrow. Later in paper, the authors distinguish these subsets as fTγδ17 (for the fetal derived Vγ4^+^ cells) and nTγδ17 (for neonatal Vγ2^+^ cells). This is awkward and confusing; why not just refer to them throughout as Vγ2^+^ Tγδ17 and Vγ4^+^ Tγδ17 instead?

Adult BM cells are inefficient in reconstituting dermal Tγδ17 (both Vγ2^+^ and Vγ4^+^) cells. Cai et al. confirmed earlier reports that thymocytes need to be co-transferred with BM cells to observe significant numbers of dermal Tγδ17 cells. No Vγ4^+^ cells arise from transferred BM cells (Figure 2D of Cai et al., 2014). This requirement for thymocytes is interpreted to support the selective ability of effector pre-programmed thymocytes to repopulate skin that lacks T cells. Overall, BM cell reconstitution studies need to be interpreted in proper contexts as Tγδ17 cells observed in chimeric mice bypass normal developmental requirements, such as neonatal thymic epithelial cells (a specific subset of medullary thymic epithelial cells, manuscript in preparation). More concerning is that the extent of Tγδ17 cell reconstitution varies from lab-to-lab (for example, Gray et al., J. Immunol, 2011; Gray et al., 2013), for reasons that are unclear. On balance, Tγδ17 cells are heterogeneous, but there is no convincing argument that dermal Tγδ17 cells are not innate programmed for function.

We do agree that using a consistent nomenclature throughout the paper will enhance clarity. As suggested, we have changed the nomenclature to be consistent throughout the paper.

3) The authors do not show that Vγ4^+^ cells remain in the dermis in Sox13^-/-^ mice, or show whether dermal Vγ4^+^ cells are actually expanded and producing IL17 in these mice. Staining to verify that dermal Vγ2^+^ cells are missing in the Sox13^-/-^ mice should also be added.

We have presented some of these data in our previous work (Malhotra et al., 2013), but we agree that these data are important for the present study. We have included the requested analysis demonstrating the loss of Vγ2^+^ cells in the skin, but normal Vγ4^+^ IL-17A expression, in new Figure 1—figure supplement 1A.

4) Regarding the AD model, the authors claim a new spontaneous atopic dermatitis model with all hallmarks of human disease. To make this claim, there should have been a more extensive characterization of the skin disease, beyond H&E staining and quantification of leukocytic infiltrate. Hallmarks of human disease that are missing (or not described?): such as itch, increased TH2 cells (IL-4/-5/-13/-31), barrier dysfunction, and increased mast cells. In addition, it would be useful to show photo images of the dermatitis in Sox13^-/-^ animals.

Please see new Figure 2D showing an increase in the number of IL-4 and/or IL-13 producing cells. We have analyzed mast cells on several occasions and found a trend toward an increase in mast cell numbers, however this did not reach statistical significance largely due to variability amongst mice with AD (new Figure 1—figure supplement 1C). We added mice scratching behavior as videos and quantitations to support an enhanced itch response (Figure 1—figure supplement 1D), as well as images of dermatitis in *Sox13^-/-^*mice (Figure 1—figure supplement 1B). An additional direct measure of barrier dysfunction, such as trans-epidermal water loss was not performed, as the acquisition and optimization of instruments required, which are not currently available at our institution, would significantly delay the resubmission beyond the specified 2 month limit.

5) Since Sox13^-/-^ mice in other backgrounds (B6) are embryonic lethal, suggesting that the animals have other developmental defects besides nTγδ17 T cell subset. The author should address whether there are other (developmental) defects in the Sox13^-/-^ mice. In addition, the authors should comment on whether keratinocytes or keratinocyte progenitors express Sox13?

Aside from the AD-like disease that develops in 129.*Sox13^-/-^*mice, the only differences between *Sox13^-/-^*and *Sox13^+/-^*LMC are the absence of Vγ2^+^ Tγδ17 cells (Malhotra et al., 2013, Spidale et al., 2018 Immunity) and reduced numbers of iNKT17 cells in the lymph nodes (Malhotra et al., 2018). iNKT17 cells, however, are rarely found in the rodent skin, as mentioned in the text. We have maintained *Sox13^-/-^*mice, of both sexes, until well over 1 year of age and have observed no other abnormalities compared to *Sox13^+/-^*mice or WT 129 mice. Regarding *Sox13* expression in keratinocytes, by RNA-seq analysis in WT keratinocytes it was not detectable (in the text), further verified by keratinocyte single cell RNAseq datasets in public domain (for e.g. https://www.biorxiv.org/content/10.1101/750042v1). We have mentioned the lack of developmental defect in the text early in the description of Figure 1. We also note the lack of *Sox13* expression in keratinocytes in the description of Figure 3.

6) The authors should de-emphasize the potential involvement of the observed clonotype (within Vβ4^+^ cells) in disease, since it is also highly abundant in non-diseased controls (Figure 6H). In addition, do CD4^+^Vβ4^+^ T cells enriched in the skin of Sox13^-/-^ mice produce IL-17 and/or IL-22 in response to skin commensals?

As alluded to by the reviewer we do not directly link the Vβ4^+^Vα4^+^ clonotype to the disease process. However, the profoundly increased cell number of this clonotype (Figure 6C) coupled with the distinct cytokine production by Vβ4^+^ cells (Figure 6D), which is a surrogate for this individual clonotype due to the fact that it represents ~75% of all Vβ4^+^ cells, is highly suggestive of a unique role for these cells in the skin. While the expansion of CD4^+^Vβ4^+^ skin T cells is dependent on skin commensals based on the antibiotic studies, our T cell hybridomas expressing the clonotype do not respond to select skin commensal species so far tested. We will need to test comprehensive sets of skin commensals to make definitive conclusions. In on-going studies we have generated the clonotypic αβTCR transgenic mice to further interrogate the unique biology underpinning this skin T cell clonotype.

7) In Figure 2—figure supplement 1H, please clarify why IL-17 is decreased in the skin in Sox13^-/-^ mice.

While we cannot definitively answer this question, we propose the following interpretation: while it is true that there is a dramatic expansion of IL-17^+^αβ T cells, there is also significant loss of IL-17 producing Vγ2^+^γδT cells, which represent a significant portion of skin leukocytes in healthy LMC skin. In addition, significant infiltration/expansion of myeloid and granulocyte populations further reduces the frequency of IL-17^+^ cells. Thus, when qPCR data is normalized on a per mg of tissue basis as done here, there is the appearance of an overall decrease in IL-17. We believe that our detailed flow cytometry immunophenotyping of IL-17 producing cells in the skin more accurately illuminates the status of IL-17 in *Sox13^-/-^*AD skin, and have emphasized these data in the main figures.